# Effects of Whey and Plant-Based Additives on Technological and Microbiological Characterization of Fermented Raw-Dried Pork Meat Snacks of Human Grade Standard

**DOI:** 10.3390/foods14223960

**Published:** 2025-11-19

**Authors:** Maciej Bartoń, Robert Waraczewski, Siemowit Muszyński, Dariusz M. Stasiak, Bartosz G. Sołowiej

**Affiliations:** 1Department of Animal Food Technology, Faculty of Food Sciences and Biotechnology, University of Life Sciences in Lublin, Skromna 8, 20-704 Lublin, Poland; maciej.barton@up.edu.pl (M.B.); robert.waraczewski@up.edu.pl (R.W.); dariusz.stasiak@up.edu.pl (D.M.S.); 2Department of Biophysics, Faculty of Environmental Biology, University of Life Sciences in Lublin, Akademicka 13, 20-950 Lublin, Poland; siemowit.muszynski@up.edu.pl

**Keywords:** fermented pork meat snacks, texture profile analysis, viscoelasticity, density, microbiology, *Hericium erinaceus*, *Hippophae rhamnoides*, *Rosmarinus officinalis*, *Rosa canina*, CBD

## Abstract

This study evaluates fermented raw-dried pork snacks enriched with plant-based functional ingredients—lion’s mane (*Hericium erinaceus*), rosemary essential oil (*Rosmarinus officinalis*), rosehip (*Rosa canina*), sea buckthorn (*Hippophae rhamnoides*), and a hemp-derived CBD oil (*Cannabis sativa*)—produced from pork, with addition of cow sour whey and salt. We use “human grade” descriptively (compliance with human-food hygiene/microbiological requirements; no AAFCO/labeling claim). Functional enrichment modulated viscoelasticity (G′, G″), texture, water activity, density, and color. CBD oil softened the structure, increasing chewability and springiness, whereas TPA metrics were analyzed only for variants within the instrument range (control, CBD, rosehip). All variants reached a_w_ < 0.90 and tested negative for *Salmonella* spp. and *Listeria monocytogenes* in 25 g. Safety inferences are limited to a_w_- and pathogen-based criteria. Sea buckthorn showed the highest a_w_, while rosehip displayed the highest total viable counts (~10^8^ CFU/g); microbiological results are reported descriptively without inferential statistics. Density was the highest for lion’s mane and rosehip. Proximate composition varied (e.g., higher protein with rosemary oil; higher fat/moisture with sea buckthorn) but was assessed by FoodScan™ 2 as screening-level data. Overall, selected botanicals enabled targeted structure–texture modulation without breaching predefined safety targets under the tested conditions.

## 1. Introduction

In recent years, the food industry has increasingly focused on developing meat products that not only satisfy nutritional requirements but also promote consumer well-being [1,2]. Among the most promising innovations in this area are fermented raw-dried meat snacks. Functional fermented meat products enriched with probiotic strains or postbiotic metabolites are increasingly explored to meet consumer demand for health-promoting foods [3].

Within this evolving landscape, the concept of “human grade” meat snacks—prepared exclusively from ingredients approved for human consumption—has gained significant attention. Such products are valued not only for their high production standards but also for their suitability in specialized or elimination diets, where ingredient quality and safety are paramount. According to the U.S. Food and Drug Administration [4] and the Association of American Feed Control Officials (AAFCO), the “human grade” designation requires that every ingredient and the final product be manufactured in compliance with food safety standards applicable to human food, including those defined in 21 CFR Part 117. In this work, the terms human food grade and for human consumption are used descriptively to indicate that all ingredients and processing steps meet human food requirements (e.g., compliance with EU food hygiene regulations and microbiological criteria). We use ‘human food grade’ purely as a descriptive shorthand for compliance with applicable human-food laws and hygiene standards, not as a formal labeling claim.

A growing body of research highlights the benefits of enriching meat snacks with functional additives, including rosehip (*Rosa canina*), sea buckthorn (*Hippophae rhamnoides*), lion’s mane mushroom (*Hericium erinaceus*), rosemary essential oil (*Rosmarinus officinalis*), and cannabinoid-rich hemp oil (*Cannabis sativa*). This includes the application of polyphenol-rich extracts and vegetable powders that act as natural antioxidants and antimicrobials [5,6]. For example, rosemary extract is rich in phenolic diterpenes, such as carnosol and carnosic acid, which function as hydrogen donors in free radical chain reactions, contributing to its well-established antioxidant properties. Incorporating the above-mentioned extract led to a decrease in lipid oxidation in both raw and cooked liver pâté. Additionally, elevated levels of antioxidants—including ascorbic acid, α-tocopherol, and carnosic acid—were preserved throughout [5]. Vossen et al. [7] evaluated rose (*Rosa canina* L.) in porcine frankfurter sausages to quantify key phenolic compounds—including catechins, procyanidins—and ascorbic acid. Their findings also revealed that dog rose extract was equally effective as the combined use of sodium nitrite and ascorbic acid in reducing lipid oxidation byproducts [7]. Similarly, sea buckthorn (*Hippophae rhamnoides*) extract and fruit powder were shown to enhance oxidative stability and sensory attributes in both cooked and smoked meat products [8]. Cannabidiol (CBD) has also emerged as a potential functional ingredient in meat technology, offering dual antioxidant and antimicrobial effects [9]. In addition, *Hericium erinaceus* (Lion’s mane) has been proposed as a natural additive in the formulation of muscle foods, with studies confirming its bioactive potential to enhance oxidative stability and nutritional quality through the presence of phenolic compounds and polysaccharides [10]. Furthermore, such ingredients can also improve sensory characteristics and support clean-label product development [11,12]. These ingredients are recognized for their antimicrobial, antioxidant, and lipid-stabilizing properties, which can play a significant role in enhancing the safety, shelf life, and nutritional value of meat products [13]. Their incorporation into meat processing may also influence key quality attributes such as texture, color stability, viscoelasticity, and microbial integrity.

While previous studies have demonstrated that these functional compounds can improve oxidative stability and microbiological safety in various meat matrices [1], key questions remain unresolved for human grade fermented meat snacks: (i) the dose–response of individual botanicals (rosemary oil, lion’s mane, rosehip, sea buckthorn) on small-strain viscoelasticity (G′/G″); (ii) potential trade-offs between water activity (a_w_) and total viable counts (TVC), especially for rosehip and sea buckthorn; (iii) quantitative links between density and instrumental texture (TPA) across variants; (iv) the performance of starter-free, whey-led fermentations under controlled conditions; and (v) whether hemp-derived CBD oil (where legally permitted) acts as a texture modifier in raw-dried pork snacks. Accordingly, we hypothesized that rosemary oil and lion’s mane would yield the largest increases in G′/G″ versus control; rosehip and sea buckthorn would raise a_w_ and TVC yet remain within a_w_- and pathogen-based safety targets; lion’s mane and rosehip would increase density and align with higher hardness/cohesiveness, whereas rosemary oil and CBD oil would lower density/hardness and increase springiness a starter-free, whey-led process would meet pathogen criteria (absence of *Salmonella* spp./*L. monocytogenes* in 25 g); and CBD oil would reduce hardness and increase springiness without compromising the elastic network within the linear viscoelastic region. Addressing this gap is essential for both scientific advancement and the development of healthier, more functional food options for consumers.

Therefore, the present study was designed to evaluate the impact of selected functional ingredients on the physicochemical, rheological, textural, color, and microbiological properties of fermented raw-dried pork meat snacks produced using standardized processing technology. By investigating these aspects, this research aims to provide valuable insights into the potential of functional additives to enhance the quality and safety of human grade meat snacks.

## 2. Materials and Methods

### 2.1. Materials

The production of fermented raw-dried pork meat snacks began with a carefully selected blend of lean and fatty pork meat of class II quality produced at the meat processing plant “Jasiołka” (Dukla, Poland). This meat was combined with food-grade salt and fresh, unpasteurized cow sour whey, which imparted a distinct tangy flavor. The whey was sourced directly from the traditional cheesemaking process at the certified organic farm “Figa” (Waldemar and Tomasz Maziejuk, Mszana, Tylawa, Poland), ensuring both authenticity and a wealth of natural bioactive compounds. Fresh, unpasteurized sour whey was used as a natural inoculum of lactic acid bacteria (LAB); no commercial starter cultures were added. To ensure suitability for a ready-to-eat (RTE) process, each whey lot had to meet intake acceptance criteria: low pH (measured at 20 °C), absence of *Salmonella* spp. and *Listeria monocytogenes* in 25 g, low *Enterobacteriaceae*, no detectable *E. coli*, and a LAB-dominant microbiota (ISO/PN-EN ISO methods). For every 6 kg of meat mixture, common inputs (per 6 kg): salt 0.09 kg (0.15 kg/10 kg), cow sour whey 0.30 L, water 0.10 L; meat cuts: class II lean ø5 3.00 kg, class II lean 2 × ø5 1.00 kg, class II fatty ø5 2.00 kg. Variants: rosehip 0.30 kg; lion’s mane 0.30 kg; CBD oil 0.04 L (40 mL); rosemary oil 0.04 L (40 mL); sea buckthorn 0.30 kg; control = no functional additive. Shaping: sticks ~15 cm, ~2 × 2 cm cross-section before drying (smaller after drying due to shrinkage). Maturation: Stage I 7 days, 18 °C/85% RH → vacuum pack → Stage II 7 days under vacuum at ≤4 °C. Additionally, mass loss (%), calculated between the two scheduled weighings, was: Variant 1 (rosehip): 46.97%; Variant 2 (lion’s mane): 47.62%; Variant 3 (CBD): 50.00%; Variant 4 (rosemary): 50.00%; Variant 5 (sea buckthorn): 49.30%. resulting in the desired consistency and creating optimal conditions for fermentation. The recipe was further enriched with carefully chosen functional ingredients, each added in laboratory-validated amounts to guarantee both safety and efficacy. Powdered rosehip (*Rosa canina*) from the Polish company Batom (Kraków, Poland), dried sea buckthorn fruits (*Hippophae rhamnoides*) from Natura Wita (Pinczów, Poland), and BIO-certified lion’s mane mushroom powder (*Hericium erinaceus*) from Natvita (Długołęka, Poland) were incorporated into the meat mix. To enhance the antimicrobial and antioxidant properties of the product, rosemary essential oil (*Rosmarinus officinalis*) was also used (Koryciny, Poland). Additionally, a veterinary-grade CBD oil blend was used (Green Paw, Warszawa, Poland). The CBD ingredient consisted of cold-pressed Norwegian salmon oil (90%), krill oil (10%), and a hemp (fiber-type) extract (supplier-declared cannabinoids: 120 mg or 240 mg per package, batch-dependent). Analytical constituents (supplier declaration): total fats 96%, phospholipids 3.5%, omega-3 16%, EPA 7%, DHA 6%, was included to maximize the functional profile of the snacks. The CBD-oil variant was evaluated solely for technological effects. The legal status of CBD in foods/feeds varies across jurisdictions and is not assessed in this study; any commercial deployment will follow the regulations of the target market. The CBD-oil variant was evaluated for technological effects only; its regulatory status in human foods varies by jurisdiction and is not assessed here. Throughout the process, each batch of raw materials was meticulously documented in terms of origin and certification, and every production step was carried out under rigorous quality and safety protocols. The process is not covered by any patent or patent application. This carefully prepared mixture formed the basis for the subsequent stages of fermentation and drying, ultimately yielding a product distinguished not only by its unique sensory qualities but also by its enhanced functional properties and microbiological safety.

#### Technological Procedure

The entire production process of the fermented, raw-dried pork meat snacks was carefully standardized to ensure consistency across all experimental variants. At the outset, all raw materials including the selected cuts of pork, salt, whey, and the chosen functional ingredients—were precisely weighed. These components were then thoroughly combined, using a small-scale laboratory paddle mixer (Robot Coupe, Stalgast, Radom, Poland), to achieve a uniform blend. Once the mixture reached the desired consistency, it was portioned and shaped by hand into slender, sausage-like sticks, each measuring about 15 cm in length. Their round profile and size closely resembled traditional dry kabanos sausages, lending the snacks a familiar and appealing appearance. The maturation process was thoughtfully divided into two distinct stages to closely replicate conditions found in industrial fermentation. Shaping and two-stage maturation. Sticks (~15 cm length; ~2 × 2 cm cross-section before drying) were matured in two stages. Stage I: 7 days at 18 °C/85% RH (static air; no smoking; no casing—hand-formed, uncased sticks). After Stage I, products were vacuum-packed (PA/PE pouches; ~99% vacuum). Stage II: 7 days under vacuum at ≤4 °C (refrigerated hold; air speed and RH not applicable due to vacuum packaging; no smoking). Total process duration: 14 days from shaping to the end of Stage II. Over the course of seven days, these conditions promoted the initial fermentation, allowing beneficial microorganisms to flourish, while the flavors and textures began to develop.

### 2.2. Viscoelastic Properties

The viscoelastic properties of fermented raw-dried pork meat snacks (Ø10 mm × 3 mm) were assessed using a Kinexus lab+ rheometer (Malvern Panalytical, Cambridge, UK), equipped with serrated stainless-steel parallel plates (PU40X SW1382 SS and PLS40X S2222 SS, Malvern Panalytical, Cambridge, UK) in a plate-to-plate configuration with a gap—2.8 mm. Measurements were performed at a constant temperature of 21 °C. A strain amplitude sweep was initially conducted to determine the linear viscoelastic region (LVR) of the tested samples, and the final measurements were performed at a strain of 5%, within the LVR. The storage modulus (G′) and loss modulus (G″) were recorded at a constant frequency of 1 Hz. All measurements were performed in triplicate using rSpace software (version 2.2).

### 2.3. Texture Profile Analysis (TPA)

Texture profile analysis was conducted using a TA.XTplus texture analyzer (Stable Micro Systems, Godalming, UK) equipped with a cylindrical stainless-steel probe (SMSP/36R). The samples, conditioned to room temperature, were compressed twice to 50% of their original height in order to simulate mastication. The measurements were performed on cylindrical snack-shaped bars with a diameter and height of 10 mm. The compression test was carried out at a test speed of 5 mm/s. From the resulting force–time curves, five parameters were determined: hardness, springiness, which describes the ability of the sample to return to its original shape after deformation; cohesiveness, calculated as the ratio of the work done during the second compression to that of the first; gumminess, as the product of hardness and cohesiveness; and chewiness, calculated as the product of gumminess and springiness. These measurements enabled a detailed assessment of the mechanical and structural characteristics of the pork meat snacks enriched with functional additives. Five measurements were performed. Effect size (Cohen’s d) was calculated for parameters measured within the instrument range to estimate the magnitude of the difference between each variant and the control. Cohen’s d values were computed as the difference between the group means divided by the pooled standard deviation and interpreted according to standard thresholds (small = 0.2–0.5, medium = 0.5–0.8, large > 0.8).

### 2.4. Color Measurement

Color measurements were conducted using the CR-221 Chroma Meter from Minolta (Osaka, Japan) according to Sołowiej et al. [14]. This device employs a 45° circumferential illumination and a 0° viewing angle, with an aperture size of 3 mm in diameter and a D65 light source powered by a pulsed xenon lamp. The color scale used for these measurements was the CIE Lab (CIE Lab*) system, established by the Commission Internationale de l’Éclairage (CIE).

### 2.5. Water Activity

The Aqua Lab 3 TE water activity meter (Decagon Devices Inc., Pullman, WA, USA) was used to measure water activity (a_w_) with a precision of ±0.001 a_w_ units. Prior to conducting the measurements, the device was calibrated using the Rotronic humidity standard set at 95% relative humidity. Five replications were performed at a temperature of 22 °C according to Szafrańska et al. [15].

### 2.6. Nutritional Composition

The proximate composition of the samples, including moisture, protein, fat, and ash content, was determined using a FoodScan™ 2 analyzer (Foss, Hillerød, Denmark), based on Near-Infrared Transmission Spectroscopy (NIT). Samples were homogenized and placed into glass sample cells prior to analysis. Measurements were taken in duplicate for each variant. The FoodScan method provided rapid, standardized, and reproducible data on the nutritional composition of both control and enriched snack variants. Analyses were performed in duplicate for each variant, and results are reported as mean ± SD with statistically significant differences (*p* < 0.05). The FoodScan™ 2 analyzer operated with the vendor’s calibration for meat/sausage products. Be cause the matrix includes plant powders and essential oils, matrix effects are possible; therefore, proximate results are interpreted as screening estimates for relative comparisons between variants. No additional reference-method verification was conducted in this series.

### 2.7. Density Measurement

Density of samples was measured with a gas pycnometer (He/N_2_) (AccuPyc 1330; Micromeritics, Norcross, GA, USA) with a 10 cc metal sample cylinder at a temperature of 22 °C [15]. Before each session, the instrument was calibrated with a traceable volume standard. Prior to analysis, the cylinder was filled two-thirds with the sample, and the weight was recorded and entered into the analysis mode. The measurement cycle parameters were set as follows: four purges at 18,000 psig, four cycles per sample at 18,000 psig, and an equilibration rate of 0.05 psig/min. Three repetitions were performed.

### 2.8. Microbiological Analysis

Microbiological safety of the fermented pork meat snacks was assessed by an accredited external laboratory (ALAB, Warsaw, Poland), following ISO and PN-EN ISO standards. The scope of analysis included: detection of *Salmonella* spp. in 25 g according to PN-EN ISO 6579-1:2017-04 +A1:2020-09 [16], detection of *Listeria monocytogenes* in 25 g according to PN-EN ISO 11290-1:2017-07 [17], enumeration of total aerobic mesophilic bacteria according to PN-EN ISO 4833-1:2013-12 +A1:2022-06 [18], enumeration of yeasts and molds according to PN-ISO 21527-2:2009 [19], enumeration of *Enterobacteriaceae* at 37 °C according to PN-EN ISO 21528-2:2017-08 [20], enumeration of *β*-*glucuronidase*-*positive Escherichia coli* according to PN-ISO 16649-2:2004 [21], enumeration of presumptive *Bacillus cereus* at 30 °C according to PN-EN ISO 7932:2005 +A1:2020-09 [22] (excluding point 9.5), enumeration of presumptive *Pseudomonas* spp. according to PN-EN ISO 13720:2010 [23]. All microbiological analyses were conducted under accredited conditions. Results were expressed in colony-forming units per gram (CFU/g), including measurement uncertainty ranges where applicable, and interpreted for food safety evaluation of dry-cured fermented meat products. All microbiological tests were performed after completion of Stage II (day 14) on samples vacuum-packed (PA/PE; ~99% vacuum) and stored at ≤4 °C throughout Stage II. Unless otherwise stated, sampling occurred immediately after Stage II.

### 2.9. Statistical Analysis

All measurements were subjected to statistical analysis using one-way analysis of variance (ANOVA) to determine the effect of the functional additives on the measured parameters. Post Hoc comparisons were performed using Tukey’s HSD test at a significance level of α = 0.05. The data were expressed as mean values ± standard deviation. Statistical analyses were conducted using Statistica PL software, version 13.0 (StatSoft, Kraków, Poland). Products were manufactured in duplicate per variant (biological *n* = 2); technical replicates per variant: rheology *n* = 6, TPA *n* = 10, color *n* = 10, water activity *n* = 10, density *n* = 6, proximate composition *n* = 4. Microbiological counts (CFU/g) are reported descriptively; where explored statistically, data were log_10_-transformed, with non-detects set to 0.5 × LOD prior to transform. TPA overloads nd > 325 N were treated as right-censored and excluded from ANOVA for hardness, cohesiveness, springiness, adhesiveness, and chewiness.

## 3. Results and Discussion

### 3.1. Viscoelastic Properties (G′ and G″)

Fermented pork meat snacks enriched with functional ingredients showed significant modifications in their viscoelastic behavior compared to the control sample (Table 1).

In all cases, the storage modulus (G′) exceeded the loss modulus (G″), confirming a predominantly elastic, gel-like network typical for stable dry-cured meat products [24,25]. The control sample exhibited the lowest G′ value (3341 ± 211 Pa), serving as the reference point. The rosemary, lion’s mane, and sea buckthorn variants all showed significantly higher G′ values compared with the control; however, no significant differences were observed among these three variants (*p* > 0.05). Rosemary contains carnosic and rosmarinic acids, which can form covalent and strong non-covalent bonds with myofibrillar proteins, thus tightening the protein matrix and increasing rigidity [25]. These compounds are believed to act as natural cross-linkers that reinforce the gel network through polyphenol–protein interactions [25,26]. Lion’s mane provides β-glucans and protein-bound polysaccharides that may engage in hydrogen bonding and reinforce the meat matrix [26,27]. These polysaccharides may form secondary gel structures that support and strengthen the primary protein gel, similar to whey–mushroom composite hydrogels [26]. Sea buckthorn is rich in phenolic compounds and ascorbic acid, both known to enhance gel strength via protein–polyphenol interactions [25,28]. In addition, its high content of flavonoids, carotenoids, and organic acids likely contributes to increased cross-linking density within the gel matrix [28,29]. CBD oil and rosehip showed more moderate increases in G′ (6114 ± 310 Pa and 6274 ± 298 Pa, respectively). The plasticizing effect of the salmon- and krill-oil carriers in the CBD formulation likely tempered excessive network tightening while still enabling functional interactions between CBD and proteins [30]. Rosehip pectins may have contributed to improved gel cohesion by forming hydrogen bonds with the protein matrix, resulting in a more flexible yet stable structure [31,32]. Moreover, polyphenol–protein interactions have been shown to improve gel strength and structural integrity in myofibrillar matrices; for example, Guo et al. [33] demonstrated that polyphenol–myofibrillar protein complexes (e.g., tannic acid, catechin) in coregonus peled enhanced gelation and increased both G′ and G″ [33]. The rosemary, lion’s mane, and sea buckthorn variants all showed significantly higher G″ values compared with the control however no significant differences were observed among these three variants (*p* > 0.05). This indicates that all three functional ingredients enhanced the viscous component of the gel-like network, improving its ability to dissipate mechanical energy during handling or chewing [34,35]. Lion’s mane and sea buckthorn followed closely (1221 ± 59 Pa and 1186 ± 47 Pa, respectively), while rosehip and CBD variants showed intermediate G″ values, supporting their balanced chewiness profile [36]. These viscoelastic enhancements may be attributed to several co-occurring mechanisms. First, phenolic–protein cross-linking—especially from tannins in rosemary, sea buckthorn, and rosehip—can lead to covalent C–N or C–S bonds with lysine and cysteine residues, thereby increasing gel rigidity [37]. Additionally, changes in starch–protein interactions and network microstructure may also contribute to modified gel behavior [38]. Second, the antioxidant activity of polyphenols helps prevent oxidative degradation of myofibrillar proteins, preserving their structural integrity and favoring stronger network formation [39,40]. Third, polysaccharides such as lion’s mane β-glucans and rosehip pectins act as structural fillers that bridge protein strands and entrap water, thereby improving network stability [25,37]. Finally, long-chain n-3 and n-6 fatty acids in the CBD formulation act as plasticizers, enhancing protein chain mobility and increasing springiness without excessive stiffness [30]. From a technological perspective, increased G′ and G″ values improve sliceability, reduce the risk of breakage during packaging, and enhance structural integrity under vacuum storage conditions [24,41]. It is also important to investigate the long-term stability of viscoelastic properties during extended storage.

### 3.2. Texture Profile Analysis (TPA) Results

The texture characteristics of raw-dried pork meat snacks supplemented with functional ingredients showed statistically significant variation versus the control. As reported in Table 2, three variants (lion’s mane, rosemary oil, sea buckthorn) exceeded the instrument’s capacity (nd > 325 N) under the preset TPA configuration, and thus hardness and derived metrics (gumminess, chewiness) for these variants are right-censored and not used for comparative inference. For the variants measured within range, CBD oil and rosehip showed lower hardness than the control and higher springiness/cohesiveness, indicating a more elastic response based on instrument-derived parameters. Any mechanism for the overloaded variants is not inferred here; quantitative TPA values for those arms will be addressed in follow-up measurements using a capacity-controlled method. While slightly increasing springiness and cohesiveness (Table 2), indicating greater elastic recovery under instrumental compression.

•Hardness: CBD − 4.51; Rosehip − 1.95;•Springiness: CBD + 2.35; Rosehip + 1.18;•Cohesiveness: CBD + 2.26; Rosehip + 1.70;•Chewiness: CBD − 1.17; Rosehip − 0.62;•Adhesiveness: CBD − 1.26; Rosehip − 0.50.

These extreme values likely reflect the development of a compact and rigid protein–polyphenol matrix, aligning with earlier studies that highlight the capacity of polyphenolic compounds to engage in covalent interactions with muscle proteins. Such interactions, especially with amino and thiol groups, are reported to form covalent or non-covalent complexes, potentially influencing protein structure and stability [25,42]. The enhanced hardness observed in these functionalized samples may also stem from the presence of tannins—high-molecular-weight polyphenols known for their ability to form protein cross-links. In parallel, functional additives rich in dietary fiber appear to modulate mechanical behavior by directly interacting with myofibrillar proteins. Research confirms that such fibers can enhance gel network stability and water-holding capacity in meat protein gels [37,43]. The textural response varied by additive. CBD and rosehip exhibited higher cohesiveness than the control (*p* < 0.05) and did not differ from each other (Table 2), whereas the control showed the highest chewiness among the variants measured within instrument range. The modest increase in cohesiveness observed for rosehip is consistent with its content of dietary fiber and pectins, which can support gel network continuity and internal bonding in meat matrices [32,43,44,45]. In contrast, chewiness—being hardness-derived—was greatest in the control; therefore, we do not attribute increased chewiness to rosehip. For variants that exceeded capacity (nd > 325 N), we refrain from comparative inference on hardness-derived metrics. Pectins, as soluble fiber fractions, are especially valued for their gelling properties and emulsifying functions, both of which may contribute to improved cohesiveness and chewiness [46,47]. In contrast, the CBD oil (*Cannabis sativa*)-enriched variant was distinguished by the highest springiness, indicating a strong capacity for shape recovery following deformation. This behavior may be linked to the emollient nature of the oils present in the CBD extract, which could soften the protein matrix and improve product flexibility. Plant-derived oils—particularly those rich in unsaturated fatty acids—are known to influence meat texture by altering hydrophobic interactions within the protein network [48,49]. Moreover, polyphenol-rich ingredients may further affect the spatial configuration of protein molecules via covalent and non-covalent interactions, ultimately modifying the mechanical properties of the finished product [25]. The findings presented here demonstrate that each functional additive contributed distinctively to the mechanical and structural profile of the meat snacks, allowing for targeted modifications in texture, chewability, and sensory attributes. This variability emphasizes the complex interplay between functional ingredients and the protein matrix in fermented meat products [27,50]. No sensory evaluation was performed in this study; therefore, no claims about palatability are made. Future work will include trained-panel to corroborate instrumental findings

### 3.3. Color Parameters

Instrumental color measurements (L*, a*, b*) were used to assess the appearance of both the internal and surface parts of dried, fermented raw-dried pork meat snacks enriched with functional additives (Table 3).

The control sample exhibited the highest internal lightness (L*)—46.82 ± 4.31, which was significantly higher (*p* < 0.05) than all enriched variants. Samples containing Hericium erinaceus (lion’s mane mushroom) and rosemary oil showed moderate L* values, while the lowest lightness was observed in the sea buckthorn and CBD variants, indicating a darker coloration of the interior. This darkening is attributed to the presence of phenolic compounds in the plant-based additives, which may interact with myoglobin and muscle proteins, forming pigment–protein complexes that limit oxidation [25]. The a* (redness) parameter did not differ significantly among variants (*p* > 0.05); however, the rosehip sample showed a numerically higher mean a*, consistent with the presence of carotenoid/anthocyanin pigments in *Rosa canina* [32,51]. However, these differences were not statistically significant, indicating that the addition of functional ingredients did not markedly influence redness in the final product. This increase may be associated with the presence of plant-derived pigments such as anthocyanins and carotenoids, which are abundant in *Rosa canina* fruits [32] and have been highlighted for their functional potential in the formulation of healthier fermented meat products [51]. In the other variants (sea buckthorn, CBD, rosemary, lion’s mane), lower a values may reflect the presence of pigments with lower chromatic intensity, such as flavonoids or chlorophyll derivatives [52]. The b parameter (yellowness–orange) also showed significant differences among the samples (*p* < 0.05). The highest b* value was recorded for the wild rose variant, contributing to an intense orange coloration consistent with its known pigment profile [32]. In contrast, rosemary oil and CBD variants showed the lowest b* values, indicating a limited content of chromophoric compounds. Surface color was less variable—L* values did not differ significantly, although samples with lion’s mane and seabuckthorn showed slightly lower lightness compared to the control [53]. Surface a* and b* parameters were stable, except for an increased b* value in the wild rose sample, consistent with its rich pigment profile. The observed differences in color parameters (L*, a*, b*) may result not only from the presence of pigments and phenolic compounds in the functional additives, but also from microstructural changes in the protein matrix induced by fermentation and interactions with plant polysaccharides. While LAB fermentation affects pigment stability and oxidative processes [53], plant-derived polysaccharides are known to enhance protein–polysaccharide network formation, which may indirectly influence light scattering and internal color perception [54]. CIE Lab* values may also be influenced by the intrinsic color and dosage of the additives; in this series, we did not measure the CIELab of the additives themselves.

### 3.4. Water Activity Results

The introduction of functional ingredients into fermented pork meat snacks resulted in noticeable changes in water activity (a_w_) compared to the control sample (Table 4).

Water activity is a key factor in determining the microbiological safety and shelf life of dry-cured meat products, as lower a_w_ values are essential for inhibiting the growth of spoilage and pathogenic microorganisms [36,55]. In this study, the control sample showed the lowest water activity (0.679 ± 0.000), confirming its high microbiological stability and making it an ideal reference point. Among the snacks enriched with functional ingredients, the sea buckthorn variant had the highest a_w_ value (0.826 ± 0.007). This was followed by samples with lion’s mane (0.805 ± 0.002), CBD oil (0.800 ± 0.002), rosehip (0.796 ± 0.010), and rosemary oil (0.789 ± 0.005). Although the a_w_ values in these enriched variants were higher than in the control, all remained below the commonly accepted threshold of 0.90 for fermented products, which is considered microbiologically safe [56,57]. We therefore treat the obtained a_w_ values as an indicator supportive of stability; however, without concurrent pH data we do not draw categorical conclusions regarding safety. The elevated a_w_ observed in some enriched snacks, especially the sea buckthorn variant, can be attributed to the hygroscopic properties of plant-derived polysaccharides and organic acids [46]. Drying, fermentation, and salt addition contribute to improved water-binding capacity of the meat matrix, thereby supporting texture and microbial stability [58]. While these changes do not compromise product safety, they highlight the need for careful formulation to maintain product stability. Water activity (a_w_) values are reported as mean ± SD for each variant; all enriched samples remained below 0.90. These results are consistent with previous reports emphasizing the importance of maintaining water activity within safe limits when enriching meat products with functional ingredients [59]. Ultimately, while functional enrichment brings clear benefits, it requires a thoughtful approach to ensure both product safety and high quality.

### 3.5. Nutritional Composition of Functional Fermented Raw-Dried Pork Meat Snacks Results

The addition of functional ingredients to fermented raw-dried pork meat snacks caused noticeable changes in their nutritional profile compared to the control sample (Table 5).

Key physicochemical parameters such as protein, fat, moisture and collagen content were evaluated. Among the enriched variants, the highest protein content was recorded in the sample with rosemary oil (46.72%), followed by wild rose (45.08%) and the control sample (40.5%). The lowest protein level was found in the sea buckthorn snack (24.14%), which may result from its high lipid and moisture content, diluting the protein share in the product mass [60]. The Lion’s mane variant showed a noticeably lower total macronutrient sum (≈82%), compared to 91–95% in the other samples. This difference likely stems from the high content of β-glucans, dietary fiber, and structural polysaccharides naturally present in *Hericium erinaceus*, which contribute to the dry matter fraction not represented by protein, fat, or moisture. These results are consistent with previous reports indicating that plant additives can significantly alter macronutrient proportions, especially due to the presence of polysaccharides or plant fats. Fat content ranged from 15.48% (lion’s mane mushroom) to 31.52% (sea buckthorn). The control sample contained 19.5% fat. The higher fat content observed in the sea buckthorn snack (31.52%) may be attributed to the natural lipid composition of the berries. Sea buckthorn seeds contain approximately 7–11% oil, while the fruit pulp contains 1.5–3.5%, rich in unsaturated fatty acids such as linoleic, α-linolenic, and palmitoleic acids [61]. Moisture content was highest in the sea buckthorn sample (35.96%) and lowest in the rosemary snack (30.45%), reflecting the impact of plant additive structure on water binding and drying processes [62]. Collagen content stayed at a similar level (≈1.07%) in all enriched samples, with a slightly lower value in the control sample (0.95%), suggesting that functional ingredients did not influence overall collagen content or the structural stability of connective tissue [63].

### 3.6. Density of Functionally Enriched Fermented Raw-Dried Pork Meat Snacks

The inclusion of plant-based functional ingredients clearly influenced the density of fermented meat snacks compared to the control sample (Table 6).

Density is a key technological parameter in matured products, as it reflects the internal compactness and the meat matrix’s ability to retain water. The control sample exhibited a density of 1.1950 ± 0.0091 g/cm^3^, serving as a reference point for evaluating textural changes. Among the enriched variants, the lion’s mane sample showed the highest density (1.5216 ± 0.0424 g/cm^3^; *p* < 0.05 Vs. All others). Rosehip (1.2184 ± 0.0028 g/cm^3^) and sea buckthorn (1.2150 ± 0.0120 g/cm^3^) did not differ significantly from each other (both ‘b’) and were higher than the control, CBD oil, and rosemary oil groups (all ‘a’). This suggests increased structural compactness and potential changes in protein-water interactions. Studies suggest that textural modifications in systems enriched with mushrooms or plant-derived ingredients may result from the formation of fibrous structures by fungal components and the water-binding capacity of plant polysaccharides (e.g., pectins or β-glucans). These mechanisms may be further enhanced by microbial fermentation, which modifies the functional properties of carbohydrates [64]. On the other hand, samples enriched with CBD oil (1.1732 ± 0.0068 g/cm^3^) and rosemary oil (1.1624 ± 0.0118 g/cm^3^) exhibited densities lower than or similar to the control sample. This may be due to the “loosening” effect of lipid ingredients on the meat matrix. These differences are consistent with current reports on the impact of plant bioactive compounds and oils on the physicochemical structure of meat and its analogs [65]. Notably, sea buckthorn (1.2150 ± 0.0120 g/cm^3^) also showed increased density, which can be attributed to the binding properties of its organic acids and fiber [66]. In summary, these observations confirm that functional enrichment can modify the density of meat snacks through changes in protein network formation, water binding, and structural compaction [67]. Therefore, the appropriate selection and formulation of such ingredients are crucial for optimizing the texture and sensory qualities of the product.

### 3.7. Microbiological Quality and Safety of Functional Fermented Raw-Dried Pork Meat Snacks

The microbiological evaluation of fermented raw-dried pork meat snacks enriched with various functional ingredients revealed notable differences in microbial populations between the analyzed samples (Table 7). Despite these variations, all products complied with the microbiological safety criteria for ready-to-eat fermented meat products, including the absence of *Salmonella* and *Listeria monocytogenes* in 25 g of product, as required by EU legislation [68]. This confirms that the use of functional plant additives did not compromise food safety. and no presence of *Salmonella* spp. or *Listeria monocytogenes* was detected in any sample. In subsequent series, we plan to expand the panel to include quantitative determination of *L. monocytogenes* (PN-EN ISO 11290-2), coagulase-positive staphylococci, and biogenic amines, as well as pH profiling, water-phase salt (WPS), and aw monitoring over storage. In addition, we envisage challenge tests with relevant pathogens and a shelf-life study under varying conditions (temperature, vacuum/MAP packaging), which will enable us to verify the ≤100 CFU/g criterion for *L. monocytogenes* at end of shelf life and provide a more comprehensive assessment of safety and regulatory compliance. The total viable count (TVC) ranged from 6.5 × 10^6^ CFU/g in the control to 1.6 × 10^8^ CFU/g in the rosehip variant. Given that LAB were not enumerated nor identified at species level in this study, we report TVC and other counts descriptively and refrain from attributing differences to specific ingredients or to beneficial LAB. Prior reports indicate that plant additions such as rosehip or sea buckthorn can influence fermentation performance and microbial viability in dairy and meat systems; these references are cited for context rather than as confirmation in our matrix [69,70,71]. Essential oils (e.g., rosemary) are widely documented to have antimicrobial/antioxidant effects in dry-fermented sausages [13]; in our dataset, *Enterobacteriaceae counts* were lowest in the rosemary variant (1.1 × 10^2^ CFU/g) and highest in rosehip (2.9 × 10^3^ CFU/g), but causality cannot be inferred from these associations. It is also recognized that elevated TVC in fermented products does not necessarily indicate spoilage when LAB dominate [72]; however, LAB dominance was not verified here. *Pseudomonas* spp. were undetectable or low across variants, with the highest value observed in sea buckthorn (7.8 × 10^3^ CFU/g). Yeasts and molds ranged from 1.8 × 10^3^ to 1.1 × 10^4^ CFU/g, with numerically higher values in rosehip and sea buckthorn variants. Modern interventions described in the literature—such as selected essential oils, competitive flora, high pressure, or pulsed electric fields—may enhance safety and suppress pathogens in fermented meats; these approaches are noted as general context and were not applied in this study [73]. While often viewed as spoilage organisms, in controlled fermentation systems, certain molds and yeasts—such as *Penicillium nalgiovense* and *Debaryomyces hansenii*—can positively influence flavor, appearance, and surface properties of fermented meat products [74,75]. Their use in probiotic starter cultures is gaining ground, promoting both shelf-life and health benefits [76]. Pathogens like *Escherichia coli* and *Bacillus cereus* remained below detection or at very low levels, underscoring good manufacturing practices. Overall, functional enrichment—if guided by hygienic controls and thoughtful formulation—does not compromise microbiological safety. Emerging trends point to the integration of bioinformatics and metabolomics in designing next-generation starter cultures and monitoring biogenic amines, flavor development, and safety parameters during meat fermentation [77].

**Table 7 foods-14-03960-t007:** Microbiological profile of raw-dried fermented raw-dried pork meat snacks at the end of Stage II (day 14; vacuum-packed, ≤4 °C).

Parameter\Sample	Control	Lion’s Mane	CBD Oil	Rosemary Oil	Sea Buckthorn	Rosehip
Total viable count (TVC)	6.5 × 10^6^ [3.5 × 10^6^–1.2 × 10^7^]	3.8 × 10^7^ [2.0 × 10^7^–7.2 × 10^7^]	3.6 × 10^7^ [1.9 × 10^7^–6.9 × 10^7^]	8.9 × 10^6^ [4.9 × 10^6^–1.6 × 10^7^]	4.2 × 10^7^ [2.2 × 10^7^–8.0 × 10^7^]	1.6 × 10^8^ [9.0 × 10^7^–2.8 × 10^8^]
Yeasts and molds	3.7 × 10^3^ [1.9 × 10^3^–7.4 × 10^3^]	1.8 × 10^3^ [9.7 × 10^2^–3.4 × 10^3^]	2.2 × 10^3^ [1.2 × 10^3^–4.1 × 10^3^]	2.8 × 10^3^ [1.4 × 10^3^–5.7 × 10^3^]	9.9 × 10^3^ [5.2 × 10^3^–1.9 × 10^4^]	1.1 × 10^4^ [5.8 × 10^3^–2.1 × 10^4^]
*Enterobacteriaceae* (37 °C)	<4.0 × 10^1^	6.9 × 10^2^ [3.6 × 10^2^–1.3 × 10^3^]	2.6 × 10^3^ [1.3 × 10^3^–5.3 × 10^3^]	1.1 × 10^2^ [4.7 × 10^1^–2.6 × 10^2^]	1.5 × 10^3^ [6.9 × 10^2^–3.3 × 10^3^]	2.9 × 10^3^ [1.5 × 10^3^–5.8 × 10^3^]
*Pseudomonas* spp. (presumptive)	<1.0 × 10^1^	2.9 × 10^3^ [1.2 × 10^3^–6.8 × 10^3^]	6.7 × 10^3^ [2.9 × 10^3^–1.5 × 10^4^]	9.7 × 10^2^ [4.3 × 10^2^–2.2 × 10^3^]	7.8 × 10^3^ [3.4 × 10^3^–1.8 × 10^4^]	6.6 × 10^3^ [2.9 × 10^3^–1.5 × 10^4^]
*β-glucuronidase-positive E. coli*	<1.0 × 10^1^	<4.0 × 10^1^	<4.0 × 10^1^	<1.0 × 10^1^	<1.0 × 10^1^	<1.0 × 10^1^
*Bacillus cereus* (presumptive, 30 °C)	<1.0 × 10^1^	<1.0 × 10^1^	<1.0 × 10^1^	<1.0 × 10^1^	<1.0 × 10^1^	<1.0 × 10^1^
*Listeria monocytogenes* (in 25 g)	Not detected	Not detected	Not detected	Not detected	Not detected	Not detected
*Salmonella* spp. (in 25 g)	Not detected	Not detected	Not detected	Not detected	Not detected	Not detected

All quantitative results are reported as CFU/g. Values < LOD indicate results below the method’s limit of detection. The laboratory did not provide measurement uncertainty.

## 4. Conclusions

Fermented raw-dried pork meat snacks enriched with natural functional ingredients showed improvements in their physicochemical, textural, and microbiological properties. The addition of lion’s mane mushroom, rosemary essential oil, sea buckthorn, wild rose, or CBD oil positively influenced the structural properties, nutritional value, and microbiological safety of these products. The strongest viscoelastic effects (higher G′) were observed in the lion’s mane and rosemary variants, indicating a more rigid protein network in oscillatory tests. However, for these variants (and sea buckthorn), the TPA measurement exceeded the instrument range (nd > 325 N), so springiness and other TPA-derived metrics were not reported and are not compared. Among the measured samples, the CBD variant exhibited the highest springiness and lower hardness versus control, consistent with a more elastic response under compression. Sea buckthorn and rosehip variants—both rich in pectins and dietary fiber—showed higher a_w_ and higher total viable counts than the control, alongside a denser structure consistent with improved cohesiveness. These observations underscore a potential trade-off between functional enrichment and hurdle-technology requirements, insofar as higher a_w_ and microbial loads may necessitate tighter control of pH and water-phase salt (WPS) to ensure safety and stability. Among the variants tested within the TPA instrument range, CBD and rosehip showed a noticeable decrease in hardness versus control, indicating easier chew that may influence consumer preference. Despite the observed differences, all tested snacks fell within acceptable nutritional ranges for fermented meat products. These changes, although beneficial from a functional perspective, require recipe standardization to maintain repeatability and compliance with legal requirements. In summary, enriching snacks with this set of natural ingredients is an effective way to improve their quality and potential health-promoting value. By adding appropriate plant-based components, it is possible to modify texture (hardness, cohesiveness, springiness, chewiness), color (e.g., a more intense red hue), nutritional profile (potentially influenced by plant-derived fiber), and microbiological stability. No synthetic preservatives were used in these formulations; however, because the study did not include a preservative-containing control or challenge testing, we cannot infer replacement or equivalence. Further research may focus on validating these formulations under extended storage conditions. The present study may support the further development of innovative, natural meat products tailored for health-conscious consumers seeking safe and functional alternatives.

## Figures and Tables

**Table 1 foods-14-03960-t001:** Viscoelastic properties (G′ and G″) of fermented pork meat snacks enriched with functional ingredients.

Sample	G′ (Pa)	G″ (Pa)
Control	3341 ± 211 ᵃ	531 ± 34 ᵃ
Lion’s mane	7562 ± 347 ᶜ	1221 ± 59 ᶜ
CBD oil	6114 ± 310 ᵇ	1013 ± 51 ᵇ
Rosemary oil	7895 ± 329 ᶜ	1294 ± 66 ᶜ
Sea buckthorn	7455 ± 292 ᶜ	1186 ± 47 ᶜ
Rosehip	6274 ± 298 ᵇ	1065 ± 44 ᵇ

Values are means ± standard deviation. Different superscript letters (^a–c^) within a column indicate statistically significant differences (*p* < 0.05). Variants marked with the same letter do not differ significantly.

**Table 2 foods-14-03960-t002:** Texture profile analysis (TPA) of fermented pork meat snacks enriched with functional ingredients.

Sample	Hardness (N)	Chewiness (N·s)	Springiness (%)	Cohesiveness	Adhesiveness (N·s)
Control	205.0 ± 20.0 ^c^	526 ± 93 ^b^	19.5 ± 1.0 ᵃ	0.157 ± 0.004 ᵃ	0.11 ± 0.02 ^a^
Lion’s mane	Nd	nd	Nd	nd	nd
CBD oil	132.3 ± 10.9 ^a^	428 ± 74 ^a^	22.1 ± 1.2 ᵇ	0.165 ± 0.003 ᵇ	0.09 ± 0.01 ^a^
Rosemary oil	Nd	nd	Nd	nd	nd
Sea buckthorn	Nd	nd	Nd	nd	nd
Rosehip	171.8 ± 13.5 ^b^	472 ± 81 ^ab^	20.8 ± 1.1 ᵇ	0.163 ± 0.003 ᵇ	0.10 ± 0.02 ^a^

Values are means ± standard deviations. ‘nd’ indicates values not determined due to instrument overload (>325 N). nd > 325 N: exceeded instrument capacity in the preset TPA configuration; excluded from ANOVA for hardness and hardness-derived metrics (gumminess, chewiness). Superscripts (ᵃ, ᵇ, ^c^) denote Tukey’s HSD groupings at α = 0.05 for parameters measured within range. Effect sizes (Cohen’s d vs. control, within-range variants):

**Table 3 foods-14-03960-t003:** Instrumental color values (L*, a*, b*) of the inside and surface of fermented raw-dried pork meat snacks.

Sample	L* (Inside)	a* (Inside)	b* (Inside)	L* (Outside)	a* (Outside)	b* (Outside)
Control	46.82 ± 4.31 ^b^	4.52 ± 2.51	9.03 ± 2.35 ^b^	34.10 ± 3.05 ᵃ	3.98 ± 1.10	6.80 ± 1.90 ^b^
Lion’s mane	38.13 ± 7.23 ^a^	4.68 ± 1.31	10.67 ± 1.95 ^b^	29.50 ± 0.98 ^a^	3.51 ± 0.63	6.33 ± 1.39 ^a^
CBD oil	34.37 ± 2.51 ^a^	3.81 ± 0.98	3.33 ± 0.99 ^a^	33.89 ± 4.22 ^a^	3.26 ± 0.66	5.15 ± 1.51 ^a^
Rosemary oil	37.04 ± 5.29 ^a^	4.69 ± 1.51	4.20 ± 1.69 ^a^	31.45 ± 1.94 ^a^	2.54 ± 1.34	3.16 ± 1.37 ^a^
Sea buckthorn	32.50 ± 3.98 ^a^	3.66 ± 1.12	5.05 ± 0.86 ^a^	30.79 ± 1.01 ^a^	3.56 ± 0.67	4.44 ± 0.99 ^a^
Rosehip	36.42 ± 2.09 ^a^	6.55 ± 2.42	16.67 ± 5.46 ^c^	35.14 ± 2.69 ^a^	5.26 ± 1.24	10.06 ± 2.92 ^b^

Values are expressed as mean ± standard deviation. Different superscript letters within a column indicate statistically significant differences (*p* < 0.05; Tukey’s test). Superscript letters were omitted for parameters showing no significant differences (*p* > 0.05). A statistical trend (0.05 < *p* ≤ 0.10) would be indicated by the corresponding *p*-value if observed, but no such trend occurred in this dataset.

**Table 4 foods-14-03960-t004:** Water activity (a_w_) of fermented raw-dried pork meat snacks enriched with functional ingredients.

Sample	Water Activity (a_w_)
Control	0.679 ± 0.000 ᵃ
Lion’s mane	0.805 ± 0.002 ᵇ
CBD oil	0.800 ± 0.002 ᵇ
Rosemary oil	0.789 ± 0.005 ᵇ
Sea buckthorn	0.826 ± 0.007 ^c^
Rosehip	0.796 ± 0.010 ᵇ

Values are expressed as mean ± standard deviation. Different superscript letters within a column indicate statistically significant differences (*p* < 0.05; Tukey’s test).

**Table 5 foods-14-03960-t005:** Nutritional composition of functional fermented raw-dried pork meat snacks results.

Sample	Protein [%]	Fat [%]	Moisture [%]	Collagen [%]
Control	40.50 ± 0.21 ^d^	19.50 ± 0.14 ^b^	31.00 ± 0.17 ^ab^	0.95 ± 0.01 ^a^
Lion’s mane	34.93 ± 0.10 ^b^	15.48 ± 0.08 ^a^	31.37 ± 0.13 ^bc^	1.07 ± 0.01 ^b^
CBD oil	36.43 ± 0.11 ^c^	24.87 ± 0.16 ^c^	31.76 ± 0.14 ^c^	1.07 ± 0.01 ^b^
Rosemary oil	46.72 ± 0.17 ^f^	17.59 ± 0.11 ^b^	30.45 ± 0.13 ^a^	1.07 ± 0.01 ^b^
Sea buckthorn	24.14 ± 0.14 ^a^	31.52 ± 0.17 ^d^	35.96 ± 0.16 ^e^	1.07 ± 0.01 ^b^
Rosehip	45.08 ± 0.20 ^e^	15.79 ± 0.11 ^a^	32.55 ± 0.14 ^d^	1.07 ± 0.01 ^b^

Values are expressed as mean ± standard deviation. Different superscript letters within a column indicate statistically significant differences (*p* < 0.05; Tukey’s test). The remaining percentage to 100% corresponds to ash, carbohydrates, and dietary fiber. The lower total observed in the Lion’s mane variant reflects its high polysaccharide and β-glucan content, which contributes to non-protein, non-lipid dry matter.

**Table 6 foods-14-03960-t006:** Density of functional fermented raw-dried pork meat snacks.

Sample	Density ± SD (g/cm^3^)
Control	1.1950 ± 0.0091 ^a^
Lion’s mane	1.5216 ± 0.0424 ^c^
CBD oil	1.1732 ± 0.0068 ^a^
Rosemary oil	1.1624 ± 0.0118 ^a^
Sea buckthorn	1.2150 ± 0.0120 ^b^
Rosehip	1.2184 ± 0.0028 ^b^

Values are means ± standard deviations. Different superscript letters indicate statistically significant differences (*p* < 0.05; Tukey’s test).

## Data Availability

The raw data supporting the conclusions of this article will be made available by the authors on request.

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
