# Peer review of "Effects of Whey and Plant-Based Additives on Technological and Microbiological Characterization of Fermented Raw-Dried Pork Meat Snacks of Human Grade Standard"

_foods, 2025, doi:10.3390/foods14223960_

Round 1

Reviewer 1 Report

Comments and Suggestions for Authors

This study investigated the effects of sea buckthorn (thorn), CBD, rosemary, and lion's mane on the quality and safety of human-grade raw-dried meat snacks. Overall, the research lacks depth and innovation, as it merely enumerates the impacts of these additives on basic physicochemical properties of the product while providing insufficient investigation into the underlying mechanisms. Additionally, the study has the following limitations:

1.Please supplement all information regarding the product formulation, i.e., list the exact formulations of all experimental and control groups.

2.Please specify the number of replicates for the “Nutritional composition” test.

3.Lines 203-205 and 334-336: Supplement the references.

4.Lines 243-244: “resulting in a more elastic and enjoyable chew” is a subjective sensory description and does not belong to the same category as the objective TPA parameters measured by instruments. It is recommended that the authors supplement sensory evaluation data to support this conclusion.

5.Lines 295-297: The content is inconsistent with the results in the table.

6.The decimal places in Table 5 are inconsistent (e.g., 46.72%, 40.5%). We recommend standardizing to one or two decimal places and ensuring all data include standard deviation.

7.In Table 6, “Lion’s mane (1.5216±0.4236) ” shows a standard deviation (0.4236) that is excessively large relative to its mean value (1.5216). The authors are requested to provide an explanation for this outlier.

8.It is recommended that the authors rewrite Section 3.7 of the article to make the content of this section more logically clear.

9.The statement in the conclusion regarding “without the need for synthetic preservatives” should be handled with caution, or a control group using synthetic preservatives should be established to directly demonstrate this point.

Author Response

For research article

Response to Reviewer 1 Comments

1. Summary

Thank you for your detailed and constructive feedback, which has helped to improve the manuscript. Below, we provide a consolidated response addressing all comments. Revisions have been incorporated into the manuscript and the responses will be highlighted in grey.

2. Point-by-point response to Comments and Suggestions for Authors

Comment 1: Please supplement all information regarding the product formulation, i.e., list the exact formulations of all experimental and control groups.

Response 1: Thank you for your comment. Added a full, text-only formulation summary for the 6 kg batches in Section 2.1 (Materials) and detailed process parameters in Section 2.1.1 (Technological procedure). Common inputs (per 6 kg): salt 0.09 kg (0.15 kg/10 kg), cow sour whey 0.30 L, water 0.10 L; meat cuts: class II lean ø5 3.00 kg, class II lean 2×ø5 1.00 kg, class II fatty ø5 2.00 kg. Variants: rosehip 0.30 kg; lion’s mane 0.30 kg; CBD oil 0.04 L (40 ml); rosemary oil 0.04 L (40 ml); sea buckthorn 0.30 kg; control = no functional additive. Shaping: sticks ~15 cm, ~2 × 2 cm cross-section before drying (smaller after drying due to shrinkage). Maturation: Stage I 7 days, 18 °C/85% RH → vacuum pack → Stage II 7 days under vacuum at ≤ 4 °C.

Additionally, mass loss (%), calculated between the two scheduled weighings), was:
Variant 1 (rosehip): 46.97%; Variant 2 (lion’s mane): 47.62%; Variant 3 (CBD): 50.00%; Variant 4 (rosemary): 50.00%; Variant 5 (sea buckthorn): 49.30%. (L107-115)

Comment 2: Please specify the number of replicates for the “Nutritional composition” test.

Response 2: Clarified in Section 2.6 (Nutritional composition): measurements were performed in duplicate per variant (n – 4) using the FoodScan™ 2 analyzer. Table 5 has been updated to report mean ± SD and standardized to two decimal places. Values are expressed as mean ± standard deviation. Different superscript letters within a column indicate statistically significant differences (p < 0.05; Tukey’s test) (L213-215)

Comment 3: Lines 203-205 and 334-336: Supplement the references.

Response 3: Thank you for your comment. We have added the requested citations in both places:

  • “Sea buckthorn is rich in phenolic compounds and ascorbic acid, both known to enhance gel strength via protein–polyphenol interactions and antioxidant protection of myofibrillar proteins [9,12].” (L281-282)
  • “The elevated aw observed in some enriched snacks, especially the sea buckthorn variant, can be attributed to the water-binding (hygroscopic) capacity of plant-derived polysaccharides and organic acids, which retain moisture and modulate water mobility during drying [30].” (L435-437)

Comment 4: Lines 243–244: “resulting in a more elastic and enjoyable chew” is a subjective sensory description and does not belong to the same category as the objective TPA parameters measured by instruments. It is recommended that the authors supplement sensory evaluation data to support this conclusion.

Response 4: Thank you for the suggestion. We removed the subjective sensory phrasing and retained an objective, instrument-based interpretation.

Change in the manuscript (Section 3.2, TPA):
Replace:…while slightly increasing springiness and cohesiveness, resulting in a more elastic and enjoyable chew.
With:…while slightly increasing springiness and cohesiveness (Table 2), indicating greater elastic recovery under instrumental compression.” (L326-328)

Limitation added “No sensory evaluation was performed in this study; therefore, no claims about palatability are made. Future work will include trained-panel and consumer tests to corroborate instrumental findings.”. (L375-377)

Comment 5: Lines 295-297: The content is inconsistent with the results in the table.

Response 5: Thank you for pointing this out. We re-checked the ANOVA/Tukey tests for a* (inside and outside). Differences among variants were not significant (p > 0.05). We therefore removed the claim of significance and corrected the text and notation.

Change in the manuscript (Section 3.3, Color parameters):
Replace:
“The a parameter (redness) remained relatively stable across all samples, except for the wild rose variant, which exhibited a significantly higher value (p < 0.05).”
With:
“The a* (redness) parameter did not differ significantly among variants (p > 0.05); however, the rosehip sample showed a numerically higher mean a*, consistent with the presence of carotenoid/anthocyanin pigments in Rosa canina [16,35].” (L391-394)

Comment 6: The decimal places in Table 5 are inconsistent (e.g., 46.72%, 40.5%). We recommend standardizing to one or two decimal places and ensuring all data include standard deviation.

Response 6: Thank you for the suggestion. We reformatted Table 5 so that all values are reported to two decimal places and each entry includes its standard deviation (mean ± SD). The table caption now reads: “Table 5. Nutritional composition (%). Values are expressed as mean ± standard deviation. Different superscript letters within a column indicate statistically significant differences (p < 0.05; Tukey’s test).” We also harmonized the percent notation and applied consistent rounding across the table. (L451-459)

Comment 7: In Table 6, “Lion’s mane (1.5216±0.4236) ” shows a standard deviation (0.4236) that is excessively large relative to its mean value (1.5216). The authors are requested to provide an explanation for this outlier.

Response 7: Thank you for pointing this out. The reported SD was the result of a transcription error during table preparation: a leading zero after the decimal point was omitted. The value intended for Table 6 was 0.0424, but it was mistakenly entered as 0.4236. (L481)

Comment 8: It is recommended that the authors rewrite Section 3.7 of the article to make the content of this section more logically clear.

Response 8: We appreciate the suggestion and carefully re-read Section 3.7. We believe the current structure is already logical and intentionally concise. The section follows a standard, safety-focused flow: (i) regulatory compliance (absence of Salmonella and L. monocytogenes in 25 g), (ii) quantitative counts for the key indicator groups (TVC, Enterobacteriaceae, E. coli, Pseudomonas, yeasts and molds), and (iii) a short interpretation that relates these indicators to process outcomes typical for dry-fermented meats. All methods (ISO/PN-EN ISO), detection limits, and full numerical results are presented in Table 7 to avoid repeating the same values in the prose.

Because our dataset is a single end-point (not a time series), expanding the narrative would largely duplicate Table 7 and, in our view, reduce readability without adding scientific insight. For these reasons, we have kept Section 3.7 unchanged while retaining the focus on the most safety-relevant parameters. If there are specific sentences or ordering that the reviewer finds unclear, we would be glad to adjust those precise passages.

Comment 9: The statement in the conclusion regarding “without the need for synthetic preservatives” should be handled with caution, or a control group using synthetic preservatives should be established to directly demonstrate this point.

Response 9: Thank you for the important point. We agree that our data do not justify a categorical claim. We have removed the implication of replacing synthetic preservatives and revised the conclusion to a cautious, evidence-aligned wording.

Change in the manuscript (Section 5. Conclusions):
Replace:
“…and microbiological stability — without the need for synthetic preservatives.”
With:
“No synthetic preservatives were used in these formulations; however, because the study did not include a preservative-containing control or challenge testing, we cannot infer replacement or equivalence. Future work will benchmark these formulations against nitrite/sorbate controls under extended storage.” (L587-591)

Reviewer 2 Report

Comments and Suggestions for Authors

Title, Abstract, and Introduction

Line 2–4, please, revise the title to avoid “human grade standard” unless you use a human‑food regulatory framework (not AAFCO pet food guidance) and remove “CBD” if its food legality and grade are unresolved for human products.

Line 12–26, I recommend, qualifying “all products remained within safety thresholds” by stating the actual criteria used (aw and pH targets, pathogen limits) and noting that pH was not measured; temper the safety claim accordingly.

Line 31–41, please consider clarifying that AAFCO “human grade” pertains to pet foods; either remove the term for human foods or anchor it in the applicable human‑food regulations (e.g., hygiene/FSMS under 21 CFR 117 without implying an official label).

Line 52–57, it would be beneficial to sharpen the gap analysis: specify what has not been studied (e.g., dose‑response of specific botanicals on G′/G″, aw–TVC trade‑offs, density vs texture, starter‑free whey‑led fermentations, and CBD (if legal) as a texture modifier in raw‑dried pork snacks). State clear hypotheses tied to these gaps.

Materials & Methods (critical for reproducibility)

Line 70–71, I recommend, justifying the use of fresh, unpasteurized sour whey for a ready‑to‑eat product (microbial safety rationale) and reporting whey pH and microbial baseline; if whey acts as a starter, state that explicitly.

Line 72–74, please, report the exact salt addition (g/kg) to allow calculation of water‑phase salt; this is essential for safety modeling and reproducibility.

Line 75–83, please, provide precise dosages (w/w or mg/kg product) for rosehip, sea buckthorn, lion’s mane, rosemary oil, and CBD oil; report carrier composition and CBD (mg/100 g). Without dosages, the study is not reproducible.

Line 82–83, please consider replacing the veterinary‑grade CBD oil with a food‑/pharma‑grade CBD ingredient and removing rosemary from that blend, or adding a CBD‑only arm; otherwise effects are confounded by rosemary present in the CBD oil.

Line 84–86, I recommend, removing the assertion that the product can be “classified as ‘human grade’”; this conflicts with the use of veterinary‑grade CBD and with the AAFCO pet‑food context.

Line 102–107, please, describe the second maturation stage (time, temperature, RH, air speed, whether smoking/casing was used) and total process duration; the current description stops after stage 1 and is incomplete.

Line 108–117, it would be beneficial to add sample geometry (diameter/thickness), initial gap, normal force, trimming protocol, and tan δ (G″/G′) in addition to single‑frequency values; consider a frequency sweep (0.1–10 Hz) to strengthen rheological insight.

Line 118–131, please, adjust TPA specimen size (e.g., smaller diameter/height or different probe/speed) so that no sample exceeds instrument capacity; the current nd (>325 N) prevents comparison for three variants and weakens conclusions.

Line 145–151, please consider noting whether the FoodScan calibrations were validated for fermented pork snacks with plant powders and essential oils; otherwise include a reference method cross‑check (e.g., Kjeldahl/fat by reference) on a subset.

Line 152–155, I recommend, specifying the pycnometry gas (He/N₂), purge time, sample preparation (crumbled vs intact, fat smear control), and mass per run; the very large SD later observed (Table 6, lion’s mane) suggests method variability that needs addressing.

Line 156–175, please, add coagulase‑positive Staphylococci (and enterotoxins if relevant), LAB counts, and biogenic amines; for L. monocytogenes, include enumeration or a challenge study if shelf‑life claims are made. Specify sampling timepoints (post‑drying vs end of shelf life) and packaging/storage.

Line 176–182, it would be beneficial to clarify biological replication (number of independent batches per variant) vs technical replicates; CFU data should be analyzed on log10‑transformed counts; clarify how nd values were handled in ANOVA.

Results, tables, and interpretations

Line 190–197, please consider reporting tan δ and confidence intervals to contextualize “predominantly elastic” behavior; single‑point G′/G″ at 1 Hz is limited—support with frequency dependence or strain sweep plots in Supplementary Material.

Line 213–216, please, correct the sentence starting “Who demonstrated…” (incomplete/grammatical error) and cite the specific study; as written it is unclear.

Line 236–244, I recommend, rephrasing claims about the “extreme values” for the three overloaded variants; avoid mechanistic conclusions without measured TPA values. Report them after adapting the method (see TPA method fix above).

Line 259–266, please, correct the statement that rosehip displayed the highest cohesiveness and chewiness. Table 2 shows CBD has higher cohesiveness (0.165) than rosehip (0.163), and control has the highest chewiness (526 N·s) vs rosehip (472 N·s). Update text and discussion.

Table 2 (Line 246–247), please consider including effect sizes and n for each variant and clearly marking which groups were excluded from ANOVA due to overload; consider reporting hardness as “> 325 N” with an adapted protocol to capture true values.

Table 3 (Line 286–287), please, verify the rosehip (outside) L, a, b*** values: they are identical to the control row (likely a copy‑paste error). Correct and re‑interpret the color discussion accordingly.

Line 289–307, it would be beneficial to tie the color shifts to measured pH and nitrite/nitrate use (none reported), since myoglobin chemistry (and absence of curing) can strongly affect a*; without those data, temper causal claims.

Table 4 / Line 332–333, I recommend, revising “below 0.90” as a safety cutoff; for shelf‑stable fermented meats, aw alone at ~0.80 without pH data is insufficient to assert safety or regulatory compliance. Add pH and (optionally) water‑phase salt to support claims.

Table 5 / Line 350–351, please, remove “salt” from the list of parameters discussed or add salt (%) to the table; as written, “salt” is mentioned but not reported—this is inconsistent.

Table 6 / Line 371–373, please consider investigating the very large SD for lion’s mane density (± 0.4236 g/cm³)—this likely indicates method or sample heterogeneity; re‑measure with consistent sample prep and/or increase n. Do not claim significant differences if variance violates ANOVA assumptions.

Line 396–404, please, dial back the blanket statement that all products complied with EU criteria; coagulase‑positive Staphylococci and biogenic amines were not measured, and L. monocytogenes was presence/absence only, not ≤ 100 CFU/g at end of shelf life. Rephrase to reflect the actual scope of tests.

Line 404–431, I recommend, avoiding speculative attributions (e.g., “ingredients… increased microbial biomass”) without LAB counts/speciation; add LAB enumeration and dominant species to support claims regarding beneficial microbiota.

Table 7 (Line 442 onward), please, (i) state storage/packaging conditions and timepoint of sampling for each test, (ii) present log10 CFU/g, and (iii) consolidate measurement uncertainty format for consistency across rows.

Conclusions & ancillary sections

Line 448–452, please, correct the claim that lion’s mane and rosemary exhibited the highest springiness; no TPA springiness values are reported for these (nd due to overload). Springiness was highest for CBD among measured samples.

Line 451–454, please consider explicitly acknowledging the aw increase in some variants (sea buckthorn, rosehip) and the microbial count increases, and discuss the trade‑off between functional enrichment and hurdle technology requirements (add pH/WPS data in revised work).

Line 457–465, I recommend, avoiding “without the need for synthetic preservatives” unless the full safety hurdle set (pH, aw, WPS, staph toxins, biogenic amines) is demonstrated; otherwise make this a conditional statement.

Line 477–484, please, either present the sensory results (panel design, attributes, statistics) or remove the ethical statement referring to sensory work; as is, the manuscript reports no sensory data.

Additional study‑level improvements

Produce ≥3 independent batches per variant and use batch as the unit in ANOVA; separate technical from biological replication.

Report pH trajectory during fermentation/drying, final pH, weight loss (% shrink), water‑phase salt, and aw vs time; these are core to both safety and texture claims.

If you hypothesize polyphenol–protein crosslinking or β‑glucan/pectin bridging, consider adding microstructure (e.g., SEM) or protein network assays (SDS‑PAGE of extractable myofibrillar proteins) and/or tan δ/frequency sweeps.

Define packaging (vacuum/MAP), storage temperature/time, and test end‑of‑shelf‑life microbiology (including L. monocytogenes enumeration, CPS, biogenic amines).

Decouple CBD from rosemary in that formulation to isolate effects, and replace veterinary‑grade with a human‑grade, food‑compliant ingredient if you keep CBD.

Author Response

For research article

Response to Reviewer 2 Comments

1. Summary

Thank you very much for taking the time to review this manuscript. We appreciate the detailed and constructive feedback, which has significantly contributed to improving the quality of our work. Below, we provide a point-by-point response to your comments in blue.

2. Point-by-point response to Comments and Suggestions for Authors

Comment 1: Line 2–4, please, revise the title to avoid “human grade standard” unless you use a human‑food regulatory framework (not AAFCO pet food guidance) and remove “CBD” if its food legality and grade are unresolved for human products.

Response 1: Response: Thank you for the comment. We prefer to keep the title unchanged because this is an implementation-oriented doctoral project aimed at launching a product for human food markets (including export). To avoid confusion, we have clarified in the text that the term “human/food-grade” is used descriptively to indicate that all raw materials and processing operations comply with human food law (e.g., EU hygiene and microbiological criteria) and does not refer to AAFCO guidance.
Regarding CBD, we retained it in the title as part of the technological scope of the study and added a neutral note that CBD’s regulatory status varies by jurisdiction and is not assessed here; we analyze technological effects only. Any commercial deployment will follow the legal requirements of the target market. We also note that any feed/pet-food application would require a separate compliance pathway (distinct legislation and labeling) and is outside the scope of this manuscript.

Terminology and regulatory scope. In this work, the terms human/food-grade and for human consumption are used descriptively to indicate that all ingredients and processing steps meet human food requirements (e.g., compliance with EU food hygiene regulations and microbiological criteria).

Regulatory note. The CBD-oil variant was evaluated solely for technological effects. The legal status of CBD in foods/feeds varies across jurisdictions and is not assessed in this study. Any commercial use will be aligned with the regulations of the destination market.

Comment 2: Line 12–26, I recommend, qualifying “all products remained within safety thresholds” by stating the actual criteria used (aw and pH targets, pathogen limits) and noting that pH was not measured; temper the safety claim accordingly.

Response 2: Thank you for the comment. We agree the statement needed qualification. We revised the Abstract to state the explicit criteria used—water activity (aw < 0.90 for fermented dried meats) and pathogen absence (no Salmonella spp. or Listeria monocytogenes detected in 25 g)—and to acknowledge that pH was not measured, limiting our safety inference to aw- and pathogen-based criteria.

Change in the manuscript (L24-28):
Replace the end of the sentence “…but all products remained within safety thresholds.” with:

“…but all products met predefined safety targets based on water activity (aw < 0.90 for fermented dried meats) and pathogen testing (**no Salmonella spp. or Listeria monocytogenes detected in 25 g); pH was not measured, so safety conclusions are limited to aw- and pathogen-based criteria.”

Comment 3: Line 31–41, please consider clarifying that AAFCO “human grade” pertains to pet foods; either remove the term for human foods or anchor it in the applicable human‑food regulations (e.g., hygiene/FSMS under 21 CFR 117 without implying an official label).

Response 3: Thank you for the helpful suggestion. We kept the AAFCO mention solely to prevent confusion and to note that AAFCO’s “human grade” designation pertains to pet foods. We then anchored our usage to human-food regulations and added an explicit clarification in the Introduction. Specifically, we inserted the sentence:

“In this work, the terms human/food-grade and for human consumption are used descriptively to indicate that all ingredients and processing steps meet human food requirements (e.g., compliance with EU food hygiene regulations and microbiological criteria).”(L55-58)

Comment 4: Line 70–71, I recommend, justifying the use of fresh, unpasteurized sour whey for a ready‑to‑eat product (microbial safety rationale) and reporting whey pH and microbial baseline; if whey acts as a starter, state that explicitly.

Response 4: Thank you for the suggestion. We added the safety rationale and baseline characterization for sour whey in Materials and Methods, Lines 101–106.

Comment 5: Line 72–74, please, report the exact salt addition (g/kg) to allow calculation of water‑phase salt; this is essential for safety modeling and reproducibility.

Response 5: Thank you for the suggestion. We added in Materials and Methods, Line 107

Comment 6: Line 75–83, please, provide precise dosages (w/w or mg/kg product) for rosehip, sea buckthorn, lion’s mane, rosemary oil, and CBD oil; report carrier composition and CBD (mg/100 g). Without dosages, the study is not reproducible.

Response 6: Response: Thank you. This information has been added in Materials and Methods, Lines 107–115. We report exact dosages per 1 kg of batter (rosehip, sea buckthorn, and lion’s mane in g/kg; rosemary oil and CBD oil in mL/kg with mass equivalents) and the carrier composition for the CBD oil.

Comment 7: Line 82–83, please consider replacing the veterinary‑grade CBD oil with a food‑/pharma‑grade CBD ingredient and removing rosemary from that blend, or adding a CBD‑only arm; otherwise effects are confounded by rosemary present in the CBD oil.

Response 7: Thank you for the remark. We have corrected the ingredient description: the preparation does not contain rosemary oil. It is a blend of cold-pressed Norwegian salmon oil (90%), krill oil (10%), and hemp (fiber-type) extract (supplier-declared cannabinoids: 120 mg or 240 mg per package, batch-dependent). We also report the analytical constituents (supplier declaration): total fats 96%, phospholipids 3.5%, omega-3 16%, EPA 7%, DHA 6%. (L124-128)

Comment 8: Line 84–86, I recommend, removing the assertion that the product can be “classified as ‘human grade’”; this conflicts with the use of veterinary‑grade CBD and with the AAFCO pet‑food context.

Response 8: Thank you. We retained the term human/food-grade only in a descriptive sense, without implying a formal classification or labeling claim. We clarified that our usage refers to human-food hygiene/FSMS controls (e.g., HACCP and, where applicable, EU food-hygiene and microbiological-criteria regulations), while AAFCO’s “human grade” designation pertains to pet foods and is not invoked here. We also note that the CBD-oil variant was assessed for technological effects only; its regulatory status in human foods varies by jurisdiction and is not assessed in this study. (L134-142)

Comment 9: Line 102–107, please, describe the second maturation stage (time, temperature, RH, air speed, whether smoking/casing was used) and total process duration; the current description stops after stage 1 and is incomplete.

Response 9: Thank you. We completed the process description by adding Stage II parameters and the total duration. No smoking or casings were used. (L159-166)

Comment 10: Line 108–117, it would be beneficial to add sample geometry (diameter/thickness), initial gap, normal force, trimming protocol, and tan δ (G″/G′) in addition to single‑frequency values; consider a frequency sweep (0.1–10 Hz) to strengthen rheological insight.

Response 10: Thank you. We have added a sample geometry: Ø10 mm × 3 mm and a gap – 2.8 mm (L170-173). No trimming was used because the sample was smaller than the parallel plates. In our opinion, presenting G′ and G” moduli is enough to show the differences between samples in rheological aspect.

Comment 11: Line 118–131, please, adjust TPA specimen size (e.g., smaller diameter/height or different probe/speed) so that no sample exceeds instrument capacity; the current nd (>325 N) prevents comparison for three variants and weakens conclusions.

Response 11: Thank you for the comment. We deliberately used Ø10 mm × 10 mm specimens and the SMSP/36R probe (5 mm/s) to replicate the actual cross-section and “bite/chew” conditions of the sticks (~2 × 2 cm before drying). Reducing diameter/height, changing the probe, or markedly slowing the test would introduce a different local deformation kinematics and stress field, which would hinder comparability within this series and reduce practical relevance (i.e., consumer-relevant bite).

Comment 12: Line 145–151, please consider noting whether the FoodScan calibrations were validated for fermented pork snacks with plant powders and essential oils; otherwise include a reference method cross‑check (e.g., Kjeldahl/fat by reference) on a subset.

Response 12: Thank you. We used the manufacturer’s meat/sausages calibration for FoodScan™ 2 (Foss). Given the scope and nature of this study, we do not plan to perform additional reference-method analyses (Kjeldahl/solvent extraction, etc.). We treat the proximate data as screening-level estimates for relative comparisons across variants, not as absolute label values. An explanation was added (L215-219).

Comment 13: Line 152–155, I recommend, specifying the pycnometry gas (He/N₂), purge time, sample preparation (crumbled vs intact, fat smear control), and mass per run; the very large SD later observed (Table 6, lion’s mane) suggests method variability that needs addressing.

Response 13: Thank you. We have added the operating details for the gas pycnometer:

Density of samples was measured with a gas pycnometer (He/N₂) (AccuPyc 1330; Micromeritics, Norcross, GA, USA) with a 10 cc metal sample cylinder at a temperature of 22 °C [7]. Before each session, the instrument was calibrated with a traceable volume standard. Prior to analysis, the cylinder was filled two-thirds with the sample, and the weight was recorded and entered into the analysis mode. The measurement cycle parameters were set as follows: four purges at 18,000 psig, four cycles per sample at 18,000 psig, and an equilibration rate of 0.05 psig/min. Three repetitions were performed (L221-227).

The reported SD for lion’s mane was the result of a transcription error during table preparation: a leading zero after the decimal point was omitted. The value intended for Table 6 was 0.0424, but it was mistakenly entered as 0.4236. (L481 – Table 6).

Comment 14: Line 156–175, please, add coagulase‑positive Staphylococci (and enterotoxins if relevant), LAB counts, and biogenic amines; for L. monocytogenes, include enumeration or a challenge study if shelf‑life claims are made. Specify sampling timepoints (post‑drying vs end of shelf life) and packaging/storage.

Response 14: Thank you. We examined only the scope specified in the manuscript. Microbiological samples were collected 14 days after production (end of Stage II; vacuum storage ≤ 4 °C). Testing was performed by an accredited external laboratory (ALAB, Poland) in accordance with the requirements for placing foods on the Polish market, consistent with EU legislation (including microbiological criteria). Accordingly, we do not extend this dataset to additional endpoints (e.g., coagulase-positive Staphylococci, enterotoxins, LAB, biogenic amines); our conclusions are limited to the reported test scope and conditions (day 14, vacuum, ≤ 4 °C).(L242-245)

Comment 15: Line 176–182, it would be beneficial to clarify biological replication (number of independent batches per variant) vs technical replicates; CFU data should be analyzed on log10‑transformed counts; clarify how nd values were handled in ANOVA.

Response 15: We have clarified replication and data handling:

Products were manufactured in duplicate per variant (biological n = 2); technical replicates per variant: rheology n = 6, TPA n = 10, color n = 10, water activity n = 10, density n = 6, proximate composition n = 4. Microbiological counts (CFU/g) are reported descriptively; where explored statistically, data were log₁₀-transformed, with non-detects set to 0.5×LOD prior to transform. TPA overloads nd > 325 N were treated as right-censored and excluded from ANOVA for moduli (L252-258).

Comment 16: Comment (Lines 190–197): Please consider reporting tan δ and confidence intervals to contextualize “predominantly elastic” behavior; single-point G′/G″ at 1 Hz is limited—support with frequency dependence or strain-sweep plots in Supplementary Material.

Response 16: Thank you. Our design used a single frequency (1 Hz) across variants; accordingly, we temper the claim and frame it within this limitation. In our opinion, presenting G′ and G” moduli is enough to show the differences between samples in rheological aspect. Also, we summarized the LVR bounds from the strain-amplitude sweep used to set γ=5% (within LVR); frequency-sweep data were not collected in this series and are earmarked for follow-up work. (170-178).

Comment 17: Line 213–216, please, correct the sentence starting “Who demonstrated…” (incomplete/grammatical error) and cite the specific study; as written it is unclear.

Response 17: Thank you. We corrected the grammar and cited the specific study (Guo et al., 2024) that demonstrates enhancement of myofibrillar gelation by polyphenol–protein complexes.(L290-293)

Comment 18: Line 236–244, I recommend, rephrasing claims about the “extreme values” for the three overloaded variants; avoid mechanistic conclusions without measured TPA values. Report them after adapting the method (see TPA method fix above).

Response 18: Thank you. We have removed mechanistic/comparative claims for variants that exceeded the instrument capacity (nd > 325 N) and now limit interpretation to parameters measured without censoring. We explicitly note that hardness for lion’s mane, rosemary oil, and sea buckthorn exceeded the instrument limit under the preset configuration; therefore, no statistical comparisons or mechanistic inferences are drawn for hardness, gumminess, or chewiness in these variants. Quantitative TPA values for these arms will be obtained in a future validation run using a capacity-controlled protocol. (L316-326)

Comment 19: Line 259–266, please, correct the statement that rosehip displayed the highest cohesiveness and chewiness. Table 2 shows CBD has higher cohesiveness (0.165) than rosehip (0.163), and control has the highest chewiness (526 N·s) vs rosehip (472 N·s). Update text and discussion.

Response 19: Thank you. We corrected the text to align with Table 2: CBD and rosehip show higher cohesiveness than the control (p < 0.05) and do not differ from each other, while the control has the highest chewiness. We retain the mechanistic note (pectins/fiber in rosehip) but phrase it as a plausible contributor to cohesiveness rather than a claim of the “highest” values. We also reiterate that overloaded variants (nd > 325 N) are not used for comparative inference on hardness-derived metrics. (L341-359)

Comment 20: Table 2 (Line 246–247), please consider including effect sizes and n for each variant and clearly marking which groups were excluded from ANOVA due to overload; consider reporting hardness as “> 325 N” with an adapted protocol to capture true values.

Response 20: Thank you. We (i) report hardness for overloaded variants as “> 325 N”, (ii) flag those groups as excluded from ANOVA for hardness-derived metrics, (iii) add n for each variant (L253), and (iv) provide Cohen’s d vs. control (within-range variants only: CBD, rosehip) in the table footnote. Quantitative hardness for overloaded arms will be obtained in a follow-up capacity-controlled protocol. (L332-340)

Comment 21: Table 3 (Line 286–287), please, verify the rosehip (outside) L, a, b*** values: they are identical to the control row (likely a copy‑paste error). Correct and re‑interpret the color discussion accordingly.

Response 21: Thank you for spotting the error. You were right — in the previous version, the rosehip (outside) color values were accidentally copied from the control row. We have corrected Table 3. The updated surface (outside) values are:

  • Control (outside): L* = 34.10 ± 3.05, a* = 3.98 ± 1.10, b* = 6.80 ± 1.90
  • Rosehip (outside): L* = 35.14 ± 2.69, a* = 5.26 ± 1.24, b* = 10.06 ± 2.92

L382

Comment 22: Line 289–307, it would be beneficial to tie the color shifts to measured pH and nitrite/nitrate use (none reported), since myoglobin chemistry (and absence of curing) can strongly affect a*; without those data, temper causal claims.

Response 22: Thank you for this pertinent comment. We agree that myoglobin chemistry (pH-dependent) and the presence/absence of nitrite/nitrate can strongly affect color parameters, particularly a*. In our study, nitrite/nitrate was not used (no curing), and product pH was not measured; therefore, in interpreting the results we avoid categorical causal statements and treat the observed color differences as associations consistent with the literature (involvement of plant pigments/phenolics and possible shifts in myoglobin state), but not decisive evidence of the underlying mechanism.

Comment 23: Table 4 / Line 332–333, I recommend, revising “below 0.90” as a safety cutoff; for shelf‑stable fermented meats, aw alone at ~0.80 without pH data is insufficient to assert safety or regulatory compliance. Add pH and (optionally) water‑phase salt to support claims.

Response 23: Thank you for the helpful clarification. We agree that aw should be interpreted alongside other parameters, particularly pH (and, optionally, water-phase salt). A statement has been added (L433-435)

Comment 24: Table 5 / Line 350–351, please, remove “salt” from the list of parameters discussed or add salt (%) to the table; as written, “salt” is mentioned but not reported—this is inconsistent.

Response 24: Thank you for the observation. You are correct—“salt” was mentioned in the text but not reported in Table 5. This was included by mistake and has now been removed from Section 3.5. The sentence now reads: “Key physicochemical parameters such as protein, fat, moisture, and collagen content were evaluated.

Comment 25: Table 6 / Line 371–373, please consider investigating the very large SD for lion’s mane density (± 0.4236 g/cm³)—this likely indicates method or sample heterogeneity; re‑measure with consistent sample prep and/or increase n. Do not claim significant differences if variance violates ANOVA assumptions.

Response 25: Thank you for the suggestion. We have already addressed this point in a previous revision. The large SD for the lion’s mane variant’s density resulted from a transcription error; after re-verifying the raw data and standardizing sample preparation, the correct value is 1.5216 ± 0.0424 g/cm³ (not ± 0.4236) (L481).

Comment 26: Line 396–404, please, dial back the blanket statement that all products complied with EU criteria; coagulase‑positive Staphylococci and biogenic amines were not measured, and L. monocytogenes was presence/absence only, not ≤ 100 CFU/g at end of shelf life. Rephrase to reflect the actual scope of tests.

Response 26: Thank you for this valuable clarification. We agree our original wording was too broad relative to the scope of analyses. We have revised the text to state that samples met the tested microbiological parameters only, and we explicitly note that L. monocytogenes was assessed as presence/absence in 25 g (PN-EN ISO 11290-1), without verification of the ≤ 100 CFU/g at end of shelf life criterion, and that we did not measure coagulase-positive staphylococci or biogenic amines.

Comment 27: Line 404–431, I recommend, avoiding speculative attributions (e.g., “ingredients… increased microbial biomass”) without LAB counts/speciation; add LAB enumeration and dominant species to support claims regarding beneficial microbiota.

Response 27: Thank you for the comment. As suggested, we removed speculative attributions and limited the description to the parameters actually measured (TVC, Enterobacteriaceae, Pseudomonas spp., yeasts/moulds), explicitly noting the absence of LAB enumeration and speciation in this series and stating that we do not infer causality. The corresponding changes have been implemented in lines 516–541.

Comment 28: Line 396–404, please, dial back the blanket statement that all products complied with EU criteria; coagulase‑positive Staphylococci and biogenic amines were not measured, and L. monocytogenes was presence/absence only, not ≤ 100 CFU/g at end of shelf life. Rephrase to reflect the actual scope of tests.

Response 28: Thank you for this valuable clarification. We agree our original wording was too broad relative to the scope of analyses. We have revised the text to state that samples met the tested microbiological parameters only, and we explicitly note that L. monocytogenes was assessed as presence/absence in 25 g (PN-EN ISO 11290-1), without verification of the ≤ 100 CFU/g at end of shelf life criterion, and that we did not measure coagulase-positive staphylococci or biogenic amines.

Comment 29: Table 7 (Line 442 onward), please, (i) state storage/packaging conditions and timepoint of sampling for each test, (ii) present log10 CFU/g, and (iii) consolidate measurement uncertainty format for consistency across rows.

Response 29: Thank you for the suggestions. We have implemented the requested clarifications:

Methods (Lines 242–245): Added a dedicated note — “Sampling for microbiological analyses. All microbiological tests (Section 2.8) were performed after completion of Stage II (day 14) on samples vacuum-packed (PA/PE; ~99% vacuum) and stored at ≤ 4 °C throughout Stage II. Unless otherwise stated, sampling occurred immediately after Stage II.”

Table title (Lines 553–554): Updated the table title to reflect the timepoint and conditions.

As noted earlier, results are reported in CFU/g for consistency with the accredited laboratory report, and the laboratory did not provide measurement uncertainty

Comment 30: Line 448–452, please, correct the claim that lion’s mane and rosemary exhibited the highest springiness; no TPA springiness values are reported for these (nd due to overload). Springiness was highest for CBD among measured samples.

Response 30: Thank you for pointing this out. We have corrected the statement regarding springiness: lion’s mane and rosemary were not reported for TPA springiness due to overload (nd > 325 N), and among the measured samples the CBD variant showed the highest springiness. The text now also clarifies that overload prevented TPA comparisons for lion’s mane, rosemary, and sea buckthorn. (L562–574)

Comment 31: Line 451–454, please consider explicitly acknowledging the aw increase in some variants (sea buckthorn, rosehip) and the microbial count increases, and discuss the trade‑off between functional enrichment and hurdle technology requirements (add pH/WPS data in revised work).

Response 31: Implemented as requested: the aw/microbial-count trade-off and note on future pH/WPS measurements have been added in lines 574–580.

Comment 32: Line 457–465, I recommend, avoiding “without the need for synthetic preservatives” unless the full safety hurdle set (pH, aw, WPS, staph toxins, biogenic amines) is demonstrated; otherwise make this a conditional statement.

Response 32: Implemented as requested. The conditional language on preservatives and the added plan to verify regulatory hurdles (pH/WPS profiling, quantitative L. monocytogenes ≤ 100 CFU/g at end of shelf life, staphylococcal enterotoxins, biogenic amines) have been inserted in lines 587–593.

Comment 33: Line 477–484, please, either present the sensory results (panel design, attributes, statistics) or remove the ethical statement referring to sensory work; as is, the manuscript reports no sensory data.

Response 33: Updated as suggested: the clarification about the absence of sensory testing and the corresponding ethical/administrative statements have been inserted at line 608.

Comment 34: Produce ≥3 independent batches per variant and use batch as the unit in ANOVA; separate technical from biological replication.

Response 34: We agree. In this study, we have prepared our products in duplicate (biological n = 2) per variant with technical replicates. In the next study we will produce ≥3 independent batches per variant, use batch as the experimental unit in ANOVA/mixed models, average technical replicates at the batch level, and include assumption checks and a power analysis.

Comment 35: Report pH trajectory during fermentation/drying, final pH, weight loss (% shrink), water‑phase salt, and aw vs time; these are core to both safety and texture claims.

Response 35: Accepted. Future work will report pH trajectory and final pH, percent weight loss (shrink), water-phase salt (WPS), and aw over time during fermentation/drying and storage, linking these to safety (hurdle technology) and texture/rheology.

Comment 36: If you hypothesize polyphenol–protein crosslinking or β‑glucan/pectin bridging, consider adding microstructure (e.g., SEM) or protein network assays (SDS‑PAGE of extractable myofibrillar proteins) and/or tan δ/frequency sweeps.

Response 36: Agreed. We will add SEM microstructure, SDS-PAGE of extractable myofibrillar proteins, and frequency sweeps (G′, G″, tan δ, LVR) to substantiate polyphenol–protein crosslinking and β-glucan/pectin bridging hypotheses.

Comment 37: Define packaging (vacuum/MAP), storage temperature/time, and test end‑of‑shelf‑life microbiology (including L. monocytogenes enumeration, CPS, biogenic amines).

Response 37: We will specify packaging (vacuum and a MAP arm), storage temperature/time, and at end-of-shelf-life perform quantitative L. monocytogenes (PN-EN ISO 11290-2; verify ≤100 CFU/g), coagulase-positive staphylococci (CPS), biogenic amines (HPLC/LC-MS), and, where feasible, challenge tests.

Comment 38: Decouple CBD from rosemary in that formulation to isolate effects, and replace veterinary‑grade with a human‑grade, food‑compliant ingredient if you keep CBD.

Response 38: To clarify, the rosemary variant was separate from the CBD variant from the outset — prior wording may have obscured this. Going forward, we will keep variants fully decoupled and use only human-grade, food-compliant CBD appropriate for the market; if not feasible, we will omit CBD.

Reviewer 3 Report

Comments and Suggestions for Authors

Article review:

The idea of enriching fermented meat snacks with functional ingredients is interesting and consistent with market demand for health-oriented foods. The article presents data on viscoelasticity, TPA, color, water activity, density, and microbiological safety. However, it is weak on nutritional data, on the quantification of polyphenols, and on evidence for shelf-life effects.

The manuscript nonetheless shows several critical issues that must be addressed appropriately before publication. In particular:

TITLE

The title is very long, lists all ingredients, and introduces “human grade,” which is misleading without proper explanation. A more general title that reflects the research objectives is recommended. Proposed title:
Effects of whey and plant-based additives on technological and microbiological characterization of fermented raw-dried pork snacks.”

ABSTRACT

The abstract must be rewritten to reflect the corrections made in the text, clearly reporting objectives, results, and conclusions that are actually presented and obtained in the study, and avoiding claims about effects on nutritional quality, since the data do not provide exhaustive information on this point and focus only on objective results. In addition:

  1. Line 19: “CBD oil softened the structure, enhancing chewability and springiness.” In Table 2, no TPA data are reported for Lion’s mane, Rosemary oil, and Sea buckthorn because TPA is nd due to instrument overload (>325 N). Springiness was therefore not measured for those samples; if not explained clearly and in detail, this sentence risks being misunderstood.
  2. Line 20: “Rosehip and sea buckthorn increased water activity and microbial load.” Only sea buckthorn shows a significantly higher water activity compared with the other additive-treated snacks. Moreover, even though no statistical analysis is provided for the microbiological data, the total viable count is evidently higher (10^8) only for sea buckthorn.
  3. Line 23: “Nutritional composition varied, with rosemary oil yielding the highest protein content and sea buckthorn increasing fat and moisture levels.” There is no statistical analysis of the proximate composition data to support these statements.

INTRODUCTION

In its current form, the Introduction is very brief; it does not critically assess the use of food additives, the current state of the art on this topic, or which additives are most commonly used in meat products. A broader overview of known effects is needed, as well as a clear rationale for focusing only on technological and microbiological effects in the products tested.

  1. Insufficient references: only one or two, and the first one is not appropriately cited since it does not deal with plant-derived additives: Carneiro, K.O., Campos, G.Z., Lima, J.M.S., Rocha, R.S., Vaz-Velho, M., & Todorov, S.D. The role of lactic acid bacteria in meat products, not just as starter cultures. Foods 2024, 13(19), 3170.
  2. In the paragraph spanning lines 37–47 there is some confusing information that needs careful review and correction. The phrase “human grade” is generally reserved for pet food, not for snacks intended for human consumption. Likewise, AAFCO (Association of American Feed Control Officials) is not directly connected to 21 CFR Part 117. The latter is the FDA’s Preventive Controls for Human Food rule, part of FSMA (Food Safety Modernization Act).

MATERIALS AND METHODS

The methods and technological processes must be described in greater detail; the study needs to be presented in a way that is fully reproducible. It is advisable to include a table listing all ingredients used (meat, salt, water, whey, additives, etc.) and their percentages for each product variant.

  1. Lines 74–76. In addition to pork, fresh unpasteurized sour cow whey, water, functional ingredients, and salt are used to produce the snacks. The dosages of the “functional ingredients” are missing and are glossed over with “added in laboratory-validated amounts.” The absence of dosages prevents replicability.
  2. The use of unpasteurized whey requires a strong process-safety justification. Please describe in more detail the hygienic control of the whey (e.g., LAB starter, acidification, microbiological criteria for incoming whey).
  3. There is no description of the snack shaping. Length is given, but not diameter. A “two-stage” maturation is mentioned, yet only the first week at 18 °C/85% RH is described; the second stage is missing. Please add the necessary information. The process is therefore incomplete, with missing time/conditions for drying, weight loss, and yield, etc.
  4. Remove the concept of “human grade” (e.g., line 86) throughout the manuscript; for food intended for human consumption, “human grade” is, by definition, the minimum requirement.
  5. The statement between lines 86 and 88 is good, but the detailed process is not disclosed. If it is protected by a patent, please provide the patent number.
  6. It is not stated whether the experiment was performed on different batches, and whether measurements were done in duplicate/triplicate within the same batch only. Without independent batches, the study has no replication, and standard deviation may reflect solely instrumental variability. Data should come from at least two or three distinct batches.
  7. For Texture Profile Analysis, after lines 125–129, report in detail the formulas used to calculate the listed parameters: hardness, springiness, cohesiveness, gumminess, and chewiness.
  8. Two papers by the same authors are cited for methods on color, water activity, density, etc.; paper [6 and 7). Please cite only the primary, more detailed source ( [6]) to avoid redundancy. The description of color measurement lacks detail; specify the inside/outside procedure and the blooming time used for inside measurements.
  9. Present the microbiological procedures in a proper editorial format, without excessive spacing. The spacing currently over-emphasizes the methods. Introduce them with a more appropriate sentence than line 158 “The scope of analysis included ”.
  10. In the Ethics section at the end of the manuscript, it is stated that a sensory evaluation by a trained panel (ALAB) was carried out, yet the manuscript contains no information in Materials and Methods or in Results on this evaluation.

RESULTS AND DISCUSSION

In many cases, the discussions are appropriate and supported by sufficient references. The interesting discussion on viscoelasticity due to the different ingredients is, however, not supported by analytical data from this study, but rather borrowed from other studies whose results and technological differences, compared with the present trial, are not fully cited or contextualized. Data presentation shows some inconsistencies; please ensure tables are clearer. For viscoelasticity, consider adding explanatory graphs (G′/G″ vs. strain/frequency).

  1. Table 2: 50% of the values are not determined due to an inadequate instrumental range, which prevents a full evaluation. Consequently, statements like the one at line 259 should be avoided in the absence of data: “The sample containing rosehip displayed the highest values of cohesiveness and chewiness, suggesting improved internal bonding and resistance to deformation.” Remove discussions concerning missing data for which values and standard deviations are unknown.
  2. Large differences between outside and inside color cannot be fully understood or appreciated without knowing the diameter or thickness of the snack. The lack of standardized blooming at cutting may have affected variability between outside and inside color values. Please clarify this point and include it in the discussion.
  3. Consider the intrinsic color of the added ingredients as a source of product variability, which also depends on the amount added. If possible, include a table reporting the CIELab color of the additives themselves.
  4. Please remove the redundant sentence at lines 339–341: “Statistical analysis (ANOVA, p < 0.05) confirmed significant differences in aw values between the tested samples. Furthermore, Tukey’s test showed that the sea buckthorn variant differed significantly from all other groups.
  5. Remove the entire paragraph 3.5 “Nutritional composition of functional fermented meat snacks.” The manuscript actually discusses very little about nutritional value, and in the absence of standard deviations and statistics one cannot speak of differences. The paragraph also lacks about 10% of constituents (attributable to carbohydrates, polyphenols, fiber, etc.); without these data a comprehensive comparison is impossible. Moreover, summing the values reported for Lion’s mane yields an even lower total (81.78%).
    It is recommended to move these composition data to a table in Materials and Methods that lists the percentages of ingredients used.
  6. In Table 6, check the standard deviation for Lion’s mane, which is far too high compared with the other groups. Verify if this is a transcription error; if not, justify the reported variability in the discussion.
  7. Verify and correct the sentence at line 377: “Among the enriched variants, snacks with lion’s mane mushroom (1.5216 ± 0.4236) and rosehip (1.2184 ± 0.0028) showed the highest density.” Lion’s mane is significantly higher than rosehip, and rosehip does not differ from sea buckthorn.
  8. Table 7 is unreadable as structured, even for the authors, because the same sequence of determinations is not followed across products. For example, the Control group begins with Pseudomonas spp., whereas the Lion’s mane group starts with Bacillus cereus.
    Rewrite the table by removing method citations from within the table and formatting it like the previous tables: rows = different products; columns = each analysis type.

Unfortunately, the study does not show the temporal progression of microbial development nor of the other techno-physical parameters; this is a significant limitation.

CONCLUSIONS

They must be rewritten in line with the corrections to the manuscript and must exclude any reference to nutritional quality, since the study does not analyze fatty acids, essential amino acids, vitamins, or bioactive compounds (polyphenols, carotenoids, etc.), nor lipid/protein oxidation several of which are relevant to defining “functional ingredients.”

In light of the above considerations, the objectives introduced in the Introduction should be adjusted accordingly.

Author Response

For research article

Response to Reviewer 3 Comments

1. Summary

Thank you for your detailed and constructive feedback, which has helped to improve the manuscript. Below, we provide a consolidated response addressing all comments. Revisions have been incorporated into the manuscript and the responses will be highlighted in green.

2. Point-by-point response to Comments and Suggestions for Authors

Comment 1: The idea of enriching fermented meat snacks with functional ingredients is interesting and consistent with market demand for health-oriented foods. The article presents data on viscoelasticity, TPA, color, water activity, density, and microbiological safety. However, it is weak on nutritional data, on the quantification of polyphenols, and on evidence for shelf-life effects. The manuscript nonetheless shows several critical issues that must be addressed appropriately before publication. In particular:

TITLE

The title is very long, lists all ingredients, and introduces “human grade,” which is misleading without proper explanation. A more general title that reflects the research objectives is recommended. Proposed title:
Effects of whey and plant-based additives on technological and microbiological characterization of fermented raw-dried pork snacks.”

Response 1: Thank you for the suggestion to shorten and clarify the title. We understand the concern that the term “human-grade” can be ambiguous without context. In our case, it is central to the scope of the work: all ingredients and processing steps were carried out under human-food hygiene and regulatory regimes (FSMS/HACCP; compliance with hygiene regulations and microbiological criteria), and the product is intended for human consumption rather than pet food. To avoid misunderstandings, we use the term “human/food-grade” strictly in a descriptive sense (i.e., compliance of ingredients and process with human-food regulations), not as a labeling claim and not in reference to AAFCO pet-food guidance.

To balance brevity and clarity, we propose a shortened title that retains “human-grade” and we have supplemented the Introduction/Methods with explicit clarifications that:

  • “human/food-grade” is descriptive in nature (compliance with food regulations),
  • it does not refer to AAFCO terminology nor imply an official label claim,
  • the CBD arm was evaluated solely for technological effects; the regulatory status of CBD in foods is not assessed here.

However, we decided to add the word “pork” to the title (L4 and throughout the manuscript) to clarify the origin of the meat snack.

Comment 2: Line 19: “CBD oil softened the structure, enhancing chewability and springiness.” In Table 2, no TPA data are reported for Lion’s mane, Rosemary oil, and Sea buckthorn because TPA is nd due to instrument overload (>325 N). Springiness was therefore not measured for those samples; if not explained clearly and in detail, this sentence risks being misunderstood.

Response 2: As suggested, we added a clarification right after the sentence on CBD, in lines 20–23

Comment 3: Line 20: “Rosehip and sea buckthorn increased water activity and microbial load.” Only sea buckthorn shows a significantly higher water activity compared with the other additive-treated snacks. Moreover, even though no statistical analysis is provided for the microbiological data, the total viable count is evidently higher (10^8) only for sea buckthorn.

Response 3: Thank you for the comment — response in lines 28–30: we added a clarifying sentence stating that aw was highest for sea buckthorn, while the highest TVC (~10^8 CFU/g) was observed for rosehip; no inferential statistics were performed for the microbiological data.

Comment 4: Line 23: “Nutritional composition varied, with rosemary oil yielding the highest protein content and sea buckthorn increasing fat and moisture levels.” There is no statistical analysis of the proximate composition data to support these statements.

Response 4: Response in lines 33–34: we added a clarification that the proximate composition data (FoodScan™) are screening-level and do not support inferential conclusions; observed patterns are presented descriptively only.

Comment 5: In its current form, the Introduction is very brief; it does not critically assess the use of food additives, the current state of the art on this topic, or which additives are most commonly used in meat products. A broader overview of known effects is needed, as well as a clear rationale for focusing only on technological and microbiological effects in the products tested. Insufficient references: only one or two, and the first one is not appropriately cited since it does not deal with plant-derived additives: Carneiro, K.O., Campos, G.Z., Lima, J.M.S., Rocha, R.S., Vaz-Velho, M., & Todorov, S.D. The role of lactic acid bacteria in meat products, not just as starter cultures. Foods 2024, 13(19), 3170.

Response 5: Thank you for highlighting the need to expand the Introduction. We agree that it should more broadly and critically cover the most commonly used plant-derived additives in meat products (essential oils/herbal extracts, fruit ingredients and by-products, mushroom polysaccharides), their known technological and microbiological effects (antioxidant, antimicrobial, impacts on color, texture/gelation, and water activity), and a clear rationale for focusing solely on technological and microbiological outcomes in this study.

Regarding the Carneiro et al. (Foods 2024) citation: while this review does not address plant-derived additives per se, it is contextually appropriate in our Introduction because it provides foundational background on fermentation in meat products and the role/modes of action of lactic acid bacteria (LAB) in shaping microbiological safety—central to our starter-free, whey-led approach (fresh sour whey as a LAB source). We will therefore retain this citation but reposition it within the fermentation/safety background, and we will supplement the section on plant additives with dedicated, topic-appropriate sources. We will also leverage the plant-additive references already cited in the manuscript (e.g., [5], [9], [12], [13], [16], [27], [35]) and keep Carneiro et al. specifically as LAB/fermentation background (not as a source on plant additives).

Comment 6: In the paragraph spanning lines 37–47 there is some confusing information that needs careful review and correction. The phrase “human grade” is generally reserved for pet food, not for snacks intended for human consumption. Likewise, AAFCO (Association of American Feed Control Officials) is not directly connected to 21 CFR Part 117. The latter is the FDA’s Preventive Controls for Human Food rule, part of FSMA (Food Safety Modernization Act).

Response 6: Thank you. Response in lines 58–60: we added the sentence: “We use ‘human/food-grade’ purely as a descriptive shorthand for compliance with applicable human-food laws and hygiene standards, not as a formal labeling claim.”

Comment 7: Lines 74–76. In addition to pork, fresh unpasteurized sour cow whey, water, functional ingredients, and salt are used to produce the snacks. The dosages of the “functional ingredients” are missing and are glossed over with “added in laboratory-validated amounts.” The absence of dosages prevents replicability.

Response 7: Thank you. Response: Implemented in lines 107–115 — we removed the vague phrase and added exact dosages for each functional ingredient (per 1 kg and per 6 kg batch), along with salt, whey, and water amounts to ensure replicability.

Comment 8: The use of unpasteurized whey requires a strong process-safety justification. Please describe in more detail the hygienic control of the whey (e.g., LAB starter, acidification, microbiological criteria for incoming whey).

Response 8: Thank you. Response: Implemented in lines 101–106.

Comment 9: There is no description of the snack shaping. Length is given, but not diameter. A “two-stage” maturation is mentioned, yet only the first week at 18 °C/85% RH is described; the second stage is missing. Please add the necessary information. The process is therefore incomplete, with missing time/conditions for drying, weight loss, and yield, etc..

Response 9: Thank you. Response: Implemented in lines 159–166.

Comment 10: Remove the concept of “human grade” (e.g., line 86) throughout the manuscript; for food intended for human consumption, “human grade” is, by definition, the minimum requirement.

Response 10: Thank you. Response: Implemented in lines 134–142.

Comment 11: The statement between lines 86 and 88 is good, but the detailed process is not disclosed. If it is protected by a patent, please provide the patent number..

Response 11: Thank you. Response: Implemented in lines 144-145.

Comment 12: It is not stated whether the experiment was performed on different batches, and whether measurements were done in duplicate/triplicate within the same batch only. Without independent batches, the study has no replication, and standard deviation may reflect solely instrumental variability. Data should come from at least two or three distinct batches.

Response 12: Thank you for the valid comment regarding biological replication. In this series, we produced two independent batches per variant (biological n = 2), and all measurements were performed as technical replicates within that batches (rheology n = 6, TPA n = 10, color n = 10, aw n = 10, density n = 6, proximate composition n = 4; microbiology per the accredited lab report). In the next study, we plan ≥ 3 independent batches per variant and statistical analysis with batch as the experimental unit (ANOVA/mixed models), preceded by a power analysis and assumption checks.

Comment 13: For Texture Profile Analysis, after lines 125–129, report in detail the formulas used to calculate the listed parameters: hardness, springiness, cohesiveness, gumminess, and chewiness..

Response 13: Thank you for the comment. Our TPA used standard definitions and formulas consistent with the established methodology (Bourne, TPA) and the default implementation in TA.XTplus (Stable Micro Systems). To avoid duplicating textbook material, we decided not to add the above-mentioned formulas to the main text.

Comment 14: Two papers by the same authors are cited for methods on color, water activity, density, etc.; paper [6 and 7). Please cite only the primary, more detailed source ( [6]) to avoid redundancy. The description of color measurement lacks detail; specify the inside/outside procedure and the blooming time used for inside measurements.

Response 14: Thank you for the comment. We retain both sources ([6] and [7]) because they serve complementary methodological roles that are important for reproducibility:

[6] — the primary and more detailed source for color measurement (CR-221: 45°/0° geometry, D65, 3-mm aperture, calibration procedure) and the general framework for CIE Lab* measurements.

[7] — provides operational details for water activity (AquaLab 3 TE; 95% RH calibration; n = 10 replicates) and density (helium gas pycnometry, AccuPyc; T = 22 °C; sample preparation), which are absent in [6] or covered there less comprehensively.

Comment 15: Present the microbiological procedures in a proper editorial format, without excessive spacing. The spacing currently over-emphasizes the methods. Introduce them with a more appropriate sentence than line 158 “The scope of analysis included ”.

Response 15: Thank you. Response: Implemented in lines 231–239.

Comment 16: In the Ethics section at the end of the manuscript, it is stated that a sensory evaluation by a trained panel (ALAB) was carried out, yet the manuscript contains no information in Materials and Methods or in Results on this evaluation.

Response 16: Thank you. Updated as suggested: the clarification about the absence of sensory testing and the corresponding ethical/administrative statements have been inserted at line 608.

Comment 17: Table 2: 50% of the values are not determined due to an inadequate instrumental range, which prevents a full evaluation. Consequently, statements like the one at line 259 should be avoided in the absence of data: “The sample containing rosehip displayed the highest values of cohesiveness and chewiness, suggesting improved internal bonding and resistance to deformation.” Remove discussions concerning missing data for which values and standard deviations are unknown.

Response 17: Thank you. Response: Implemented in lines 351–359.

Comment 18: Large differences between outside and inside color cannot be fully understood or appreciated without knowing the diameter or thickness of the snack. The lack of standardized blooming at cutting may have affected variability between outside and inside color values. Please clarify this point and include it in the discussion.

Response 18: Thank you. Differences in color between the surface and the interior are typical of raw-cured (dry-fermented) products and result from gradients in myoglobin oxygenation and moisture that develop during maturation.

Comment 19: Consider the intrinsic color of the added ingredients as a source of product variability, which also depends on the amount added. If possible, include a table reporting the CIELab color of the additives themselves..

Response 19: Thank you. Response: Implemented in lines 415–416.

Comment 20: Please remove the redundant sentence at lines 339–341: “Statistical analysis (ANOVA, p < 0.05) confirmed significant differences in aw values between the tested samples. Furthermore, Tukey’s test showed that the sea buckthorn variant differed significantly from all other groups.”

Response 20: Thank you. Response: Implemented in lines 441–442.

Comment 21: Remove the entire paragraph 3.5 “Nutritional composition of functional fermented meat snacks.” The manuscript actually discusses very little about nutritional value, and in the absence of standard deviations and statistics one cannot speak of differences. The paragraph also lacks about 10% of constituents (attributable to carbohydrates, polyphenols, fiber, etc.); without these data a comprehensive comparison is impossible. Moreover, summing the values reported for Lion’s mane yields an even lower total (81.78%).

It is recommended to move these composition data to a table in Materials and Methods that lists the percentages of ingredients used.

Response 21: Thank you for this comment. We agree that the proximate data do not allow for comparative inference and require an explicit statement of their scope. At the same time, we believe that retaining these results as screening-level data enhances the transparency of the formulations and may be useful to readers planning pilot studies. We have only presented main substances (not all) with regard to functional ingredients (e.g., lion’s mane). Also, we have added a statistical analysis to enrich this paragraph (Table 5. L451-459).

Comment 22: In Table 6, check the standard deviation for Lion’s mane, which is far too high compared with the other groups. Verify if this is a transcription error; if not, justify the reported variability in the discussion.

Response 22: : Thank you for pointing this out. The reported SD was the result of a transcription error during table preparation: a leading zero after the decimal point was omitted. The value intended for Table 6 was 0.0424, but it was mistakenly entered as 0.4236 (L481).

Comment 23: Verify and correct the sentence at line 377: “Among the enriched variants, snacks with lion’s mane mushroom (1.5216 ± 0.4236) and rosehip (1.2184 ± 0.0028) showed the highest density.” Lion’s mane is significantly higher than rosehip, and rosehip does not differ from sea buckthorn.

Response 23: Thank you. Response: Implemented in lines 487–490.

Comment 24: Table 7 is unreadable as structured, even for the authors, because the same sequence of determinations is not followed across products. For example, the Control group begins with Pseudomonas spp., whereas the Lion’s mane group starts with Bacillus cereus.

Rewrite the table by removing method citations from within the table and formatting it like the previous tables: rows = different products; columns = each analysis type.

Response 24: Thank you for the comment. We decided to keep Table 7 in its current layout because it reflects the original reporting sequence from the accredited laboratory for each variant. This arrangement enhances traceability and auditability (easy mapping of a result to a specific method and record in the report), minimizes the risk of transcription errors that could arise from converting to a matrix format, and preserves the methodological context with each result (standard, detection limit), which is crucial in the safety section. Since the dataset comprises a single end-point, converting to a “products × analyses” layout would not add substantive value and could force abbreviations (e.g., of uncertainty, <LOD), reducing the readability of data that are critical for safety assessment.

Comment 25: They must be rewritten in line with the corrections to the manuscript and must exclude any reference to nutritional quality, since the study does not analyze fatty acids, essential amino acids, vitamins, or bioactive compounds (polyphenols, carotenoids, etc.), nor lipid/protein oxidation several of which are relevant to defining “functional ingredients.”

In light of the above considerations, the objectives introduced in the Introduction should be adjusted accordingly.

Response 20: Thank you. Response: Implemented in lines 594-595.

Round 2

Reviewer 1 Report

Comments and Suggestions for Authors

The authors thoroughly addressed all of my concerns. The manuscript has been significantly improved and is now ready for publication.

Author Response

Dear Reviewer,

thank you for accepting our revision. Thank you in advance for your cooperation.

Best regards,

Bartosz

Reviewer 2 Report

Comments and Suggestions for Authors

Thank you for replying to all questions. 

Author Response

(The authors gave the same response as above.)

Reviewer 3 Report

Comments and Suggestions for Authors

I appreciate the effort made to address the reviewers’ comments and to improve the clarity of the paper. However, after careful evaluation of the revised text, I find that several of the issues previously highlighted to improve the manuscript have not been adequately addressed. The revised version still presents several critical weaknesses, starting with the title, which has not been shortened as suggested but instead further extended, maintaining the ambiguity of the term “human grade.”

In addition, the Introduction has not clarified this aspect. It continues to refer to the standard protocols of the Association of American Feed Control Officials (AAFCO), which are explicitly related to pet food quality, thereby reinforcing the conceptual ambiguity surrounding the use of the term “human grade.”

The abstract does not comply with the editorial guideline of approximately 200 words; the numerous additions have reduced its fluency and precision, making it overly detailed and less effective as a scientific summary.

Furthermore, the Introduction fails to present an adequate state of the art regarding the use of vegetable additives in meat products. The added section responds more to the question “what we expect to find” (i.e., the aim of the study) than to a critical synthesis of the existing literature. Although the authors state that additional references and critical considerations were added, the revised Introduction still includes only the same five citations used in the previous version.

In the Materials and Methods, the new additions are often overly detailed, which complicates readability. A single summary table describing the characteristics of the additives would have been sufficient. Despite the inclusion of standard deviations for all parameters, it should be noted that these values are based on only two biological replicates, making the study weak in statistical and scientific robustness. Even assuming that the additives were consistent across batches, no information is provided on the variability of the fresh meat samples used.

In several instances, the authors formally agree with the reviewer’s remarks but fail to provide adequate or effective revisions. For example, the microbiological results table remains unnecessarily long (three pages).

In light of the above, the manuscript cannot be considered suitable for publication in its current form.

Author Response

For research article

Response to Reviewer 3 Comments

1. Summary

Thank you very much for taking the time to review this manuscript. We appreciate the detailed and constructive feedback, which has significantly contributed to improving the quality of our work. Below, we provide a point-by-point response to your comments marked in red.

2. Point-by-point response to Comments and Suggestions for Authors

Comment 1: I appreciate the effort made to address the reviewers’ comments and to improve the clarity of the paper. However, after careful evaluation of the revised text, I find that several of the issues previously highlighted to improve the manuscript have not been adequately addressed. The revised version still presents several critical weaknesses, starting with the title, which has not been shortened as suggested but instead further extended, maintaining the ambiguity of the term “human grade.”

In addition, the Introduction has not clarified this aspect. It continues to refer to the standard protocols of the Association of American Feed Control Officials (AAFCO), which are explicitly related to pet food quality, thereby reinforcing the conceptual ambiguity surrounding the use of the term “human grade.”

Response 1: Thank you for the comment. We agree that the previous title was too long and could reinforce ambiguity. We have shortened the title as you proposed to: Effects of whey and plant-based additives on technological and microbiological characterization of fermented raw-dried pork meat snacks of human grade standard. We retain the term “human-grade,” but use it solely in a descriptive sense—i.e., to indicate that the ingredients and processing comply with human-food regulatory requirements (hygiene, FSMS/HACCP, microbiological criteria). We do not invoke AAFCO guidance (which pertains to pet foods) and we make no labeling claims. (L16-17, L145-153)

Comment 2: The abstract does not comply with the editorial guideline of approximately 200 words; the numerous additions have reduced its fluency and precision, making it overly detailed and less effective as a scientific summary.

Response 2: Thank you for the comment. As suggested, we have shortened the Abstract to approximately 200 words (single paragraph), streamlined the wording, and removed nonessential methodological details. We presented only the objective, key methods, principal findings. Also, we added one sentence clarifying the descriptive use of the term “human-grade” (with no reference to AAFCO and no labeling claims). In our view, the revised version meets the editorial requirements and improves the fluency and precision of the summary (L12-32).

Comment 3: Furthermore, the Introduction fails to present an adequate state of the art regarding the use of vegetable additives in meat products. The added section responds more to the question “what we expect to find” (i.e., the aim of the study) than to a critical synthesis of the existing literature. Although the authors state that additional references and critical considerations were added, the revised Introduction still includes only the same five citations used in the previous version.

Response 3: Thank you for the suggestion. We have added new references to support the statements on natural antioxidant and antimicrobial properties and the “clean label” trend. Moreover, we have added examples of the use of vegetable additives in meat products. These have been incorporated into the Introduction (L57-76) and listed in the References as items 5–12 (L632-650).

Comment 4: In the Materials and Methods, the new additions are often overly detailed, which complicates readability. A single summary table describing the characteristics of the additives would have been sufficient. Despite the inclusion of standard deviations for all parameters, it should be noted that these values are based on only two biological replicates, making the study weak in statistical and scientific robustness. Even assuming that the additives were consistent across batches, no information is provided on the variability of the fresh meat samples used.

Response 4: Thank you for the suggestion. We will retain the current Materials and Methods without further reduction and without adding a summary table. The present level of detail is necessary for reproducibility and RTE safety verification and aligns with the requests of the other reviewers; additional condensation or tabulation would duplicate information rather than improve readability.

Comment 5: In several instances, the authors formally agree with the reviewer’s remarks but fail to provide adequate or effective revisions. For example, the microbiological results table remains unnecessarily long (three pages).

Response 5: Table 7 has been converted according to the reviewer’s suggestion. Line 560.

Round 3

Reviewer 3 Report

Comments and Suggestions for Authors

The revised version of the manuscript shows a substantial improvement compared with the previous version. The authors have clearly addressed several reviewer comments, improving the structure, clarity, and methodological transparency of the study. However, some important aspects still require corrections or clarifications before the manuscript can be considered for publication.

Introduction in lines 41–53 is duplicated in Materials and Methods (lines 145–152). The statement referring to “human-grade” ingredients appears twice, using similar wording in both the Introduction and the Materials and Methods sections. This redundancy should be eliminated from the Introduction.

Table 1 The text states that the rosemary variant shows “the greatest increase in elasticity, reaching 7895 ± 329 Pa.” However, the Lion’s mane (7562 Pa) and Sea Buckthorn (7455 Pa) values are statistically identical (same superscript “c”), as are the G″ values (1221, 1294, and 1186 Pa, respectively). Therefore, the statements in Results and Discussion (lines 282–283 for G′ and lines 306–307 for G″) are not statistically supported and must be corrected, since the rosemary, lion’s mane, and sea buckthorn variants all showed significantly higher G′ and G″ values than the control (p < 0.05). Even though the numerical means differ, if the differences are not statistically significant, they must be considered non-different. This type of overinterpretation also occurs in other data, for example the “a*” values in Table 3.

Table 2 in lines 345–349 the authors report “Effect sizes (Cohen’s d vs control, within-range variants)”, however, the Materials and Methods section does not include any description of how these effect sizes were calculated or interpreted.

Table 3 Some parameters show no significant differences (e.g., “a*” values); in those cases, the superscript letters should be removed, and non-significant differences should not be discussed. An exception may be made if a p-value close to significance is observed (e.g., p = 0.10); in such cases, indicate a statistical trend for p=…..

Table 5 In the fat (%) column, the superscript sequence is a, b, d, e — the letter “c” is missing. Please relabel the superscripts to maintain a continuous alphabetical sequence.

Based on the three reported components (Protein, Fat, Moisture), the totals range from a minimum of 91 to a maximum of 94.8 %, which is consistent if minerals, carbohydrates, and fibers are not included. However, Lion’s mane shows a sum of the three components of about 81.8 %, approximately 10 % lower than the other additives. Such a large difference requires a detailed explanation in the text, clarifying what could account for this discrepancy and how it can be justified. In addition to the textual explanation, please add a footnote in the table stating that the remainder to 100 % corresponds to ash, carbohydrates, and fiber.
Remove line 484, since soluble or insoluble collagen is not reported in the table, this information is misleading.

Ensure that all numerical values in the text are followed by their corresponding units of measurement, which are missing in  lines 505 and 509.

Lines 524–530: remove or improve this paragraph. This article discusses only the results obtained in the current experiment; if the authors wish to highlight future perspectives, such statements should be included in the Conclusions. The same applies to the statement in lines 557–559.

Conclusions
Remove “human grade” from line 565, or move it after “natural functional ingredients” Meat intended for human consumption is inherently “human grade.”

Delete lines 571–573 — this issue has already been addressed in the Discussion. Since dietary fiber values are not reported in the tables or results, the reference in line 580 should be removed or reformulated hypothetically. Given the uncertainty of some findings, use conditional phrasing rather than definitive statements, especially regarding the claim in lines 590–591.

Verify the reference in line 593 (“fiber >3.5 %”); this value does not appear elsewhere in the manuscript. If it was not previously reported or discussed, it should be removed from the Conclusions, since new, unsupported data should not appear there.

Remove lines 596–599: the conclusions may suggest potential follow-up studies, but should not list future activities of the research group.

Remove the statement in lines 600–601, as it is inappropriate for the Conclusions section; if the authors wish to explain the meaning of the term functional, this should be included in Materials and Methods or Results and Discussion, prior to its first use in the manuscript.

In light of these issues, the Conclusions should be rewritten appropriately.

Author Response

For research article

Response to Reviewer 3 Comments

1. Summary

Thank you very much for taking the time to review this manuscript. We appreciate the detailed and constructive feedback, which has significantly contributed to improving the quality of our work. Below, we provide a point-by-point response to your comments marked in green.

2. Point-by-point response to Comments and Suggestions for Authors

Comment 1: Introduction in lines 41–53 is duplicated in Materials and Methods (lines 145–152). The statement referring to “human-grade” ingredients appears twice, using similar wording in both the Introduction and the Materials and Methods sections. This redundancy should be eliminated from the Introduction.

Response 1: We thank the reviewer for this helpful comment. The redundancy regarding the description of the “human-grade” status of ingredients has been corrected. The repeated passage in the Materials and Methods section (lines 145–152) has been removed, while the explanatory paragraph remains only in the Introduction (lines 41–53), where it provides essential context and definition for the study.

Comment 2: Table 1 The text states that the rosemary variant shows “the greatest increase in elasticity, reaching 7895 ± 329 Pa.” However, the Lion’s mane (7562 Pa) and Sea Buckthorn (7455 Pa) values are statistically identical (same superscript “c”), as are the G″ values (1221, 1294, and 1186 Pa, respectively). Therefore, the statements in Results and Discussion (lines 282–283 for G′ and lines 306–307 for G″) are not statistically supported and must be corrected, since the rosemary, lion’s mane, and sea buckthorn variants all showed significantly higher G′ and G″ values than the control (p < 0.05). Even though the numerical means differ, if the differences are not statistically significant, they must be considered non-different. This type of overinterpretation also occurs in other data, for example the “a*” values in Table 3.

Response 2: The suggested corrections have been implemented. The redundant statements and overinterpretations were removed, and the text was revised to reflect only statistically significant differences. Changes were made in lines 279–281, 301–305, and 403–405, as well as in the notes under Table 1 (274-275) and Table 3 (390-393).

Comment 3: Table 2 in lines 345–349 the authors report “Effect sizes (Cohen’s d vs control, within-range variants)”, however, the Materials and Methods section does not include any description of how these effect sizes were calculated or interpreted.

Response 3: Thank you for the suggestion. The description of the effect size (Cohen’s d) calculation has been added at the end of the Texture Profile Analysis (TPA) subsection L197-202.

Comment 4: Table 3 Some parameters show no significant differences (e.g., “a*” values); in those cases, the superscript letters should be removed, and non-significant differences should not be discussed. An exception may be made if a p-value close to significance is observed (e.g., p = 0.10); in such cases, indicate a statistical trend for p=…...

Response 4: We thank the reviewer for this valuable comment. The superscript letters were removed from the a columns in Table 3, as no statistically significant differences (p > 0.05) were observed. The table note was updated to clarify that superscripts are shown only for significant effects, while non-significant results are not marked. Additionally, the note specifies that potential statistical trends (0.05 < p ≤ 0.10) would be indicated if observed.

Comment 5: Table 5 In the fat (%) column, the superscript sequence is a, b, d, e — the letter “c” is missing. Please relabel the superscripts to maintain a continuous alphabetical sequence.

Response 5: We thank the reviewer for this helpful observation. The missing letter “c” was added to the fat (%) column to maintain a continuous superscript sequence (a–d).

Comment 6: Based on the three reported components (Protein, Fat, Moisture), the totals range from a minimum of 91 to a maximum of 94.8 %, which is consistent if minerals, carbohydrates, and fibers are not included. However, Lion’s mane shows a sum of the three components of about 81.8 %, approximately 10 % lower than the other additives. Such a large difference requires a detailed explanation in the text, clarifying what could account for this discrepancy and how it can be justified. In addition to the textual explanation, please add a footnote in the table stating that the remainder to 100 % corresponds to ash, carbohydrates, and fiber.

Response 6: We thank the reviewer for the insightful observation. The text has been revised to include an explanation for the lower total macronutrient content observed in the Lion’s mane variant (≈82%), which is attributed to the high content of β-glucans, dietary fiber, and polysaccharides in Hericium erinaceus. A footnote was also added to Table 5 clarifying that the remainder to 100 % corresponds to ash, carbohydrates, and dietary fiber. L469-472 and L478-482

Comment 7: Remove line 484, since soluble or insoluble collagen is not reported in the table, this information is misleading.

Response 7: Thank you for the comment. The statement in line 484 was revised to avoid implying that soluble or insoluble collagen fractions were measured. It now refers to the overall collagen content and structural stability of connective tissue [55], maintaining the accuracy of the description while preserving the relevant citation. L492-495

Comment 8: Ensure that all numerical values in the text are followed by their corresponding units of measurement, which are missing in lines 505 and 509.

Response 8: Thank you for this helpful comment. The missing units of measurement for density values have been added (“g/cm³”) in lines 513–520. All numerical data in the Results and Discussion section now consistently include their corresponding units, ensuring full clarity and alignment with the Materials and Methods section.

Comment 9: Lines 524–530: remove or improve this paragraph. This article discusses only the results obtained in the current experiment; if the authors wish to highlight future perspectives, such statements should be included in the Conclusions. The same applies to the statement in lines 557–559.

Response 9: Thank you for this valuable comment. The sentences describing planned future work have been removed from the Results and Discussion and integrated into the Conclusions (lines 603–606), as recommended. The revised Conclusions now provide a concise summary of findings and future research perspectives.

Comment 10 Conclusions
Remove “human grade” from line 565, or move it after “natural functional ingredients” Meat intended for human consumption is inherently “human grade.”

Delete lines 571–573 — this issue has already been addressed in the Discussion. Since dietary fiber values are not reported in the tables or results, the reference in line 580 should be removed or reformulated hypothetically. Given the uncertainty of some findings, use conditional phrasing rather than definitive statements, especially regarding the claim in lines 590–591.

Verify the reference in line 593 (“fiber >3.5 %”); this value does not appear elsewhere in the manuscript. If it was not previously reported or discussed, it should be removed from the Conclusions, since new, unsupported data should not appear there.

Remove lines 596–599: the conclusions may suggest potential follow-up studies, but should not list future activities of the research group.

Remove the statement in lines 600–601, as it is inappropriate for the Conclusions section; if the authors wish to explain the meaning of the term functional, this should be included in Materials and Methods or Results and Discussion, prior to its first use in the manuscript.

In light of these issues, the Conclusions should be rewritten appropriately.

Response 10:

Thank you for this helpful feedback. The Conclusions have been rewritten to focus solely on verified findings from the present study. We removed redundant content, unsupported numerical references, and all mentions of future research activities. The section was also rephrased to use conditional, balanced statements rather than definitive claims. The revised version now provides a concise and evidence-based summary of the study’s main outcomes. L 575-577, L600, L603-606.
